# The identification of dual protective agents against cisplatin-induced oto- and nephrotoxicity using the zebrafish model

Jaime N Wertman[1,2], Nicole Melong[2,3], Matthew R Stoyek[4], Olivia Piccolo[2,5], Stewart Langley[2], Benno Orr[6], Shelby L Steele[7], Babak Razaghi[8], Jason N Berman[2,3]*

[1]Dalhousie University, Department of Microbiology and Immunology, Halifax, Canada; [2]IWK Health Centre, Department of Pediatrics, Halifax, Canada; [3]CHEO Research Institute, Ottawa, Canada; [4]Dalhousie University, Department of Physiology & Biophysics, Halifax, Canada; [5]McMaster University, Department of Global Health, Hamilton, Canada; [6]University of Toronto, Department of Molecular Genetics, Toronto, Canada; [7]Appili Therapeutics Inc, Halifax, Canada; [8]Dalhousie University, Faculty of Dentistry, Halifax, Canada

**Abstract** Dose-limiting toxicities for cisplatin administration, including ototoxicity and nephrotoxicity, impact the clinical utility of this effective chemotherapy agent and lead to lifelong complications, particularly in pediatric cancer survivors. Using a two-pronged drug screen employing the zebrafish lateral line as an in vivo readout for ototoxicity and kidney cell-based nephrotoxicity assay, we screened 1280 compounds and identified 22 that were both oto- and nephroprotective. Of these, dopamine and L-mimosine, a plant-based amino acid active in the dopamine pathway, were further investigated. Dopamine and L-mimosine protected the hair cells in the zebrafish otic vesicle from cisplatin-induced damage and preserved zebrafish larval glomerular filtration. Importantly, these compounds did not abrogate the cytotoxic effects of cisplatin on human cancer cells. This study provides insights into the mechanisms underlying cisplatin-induced oto- and nephrotoxicity and compelling preclinical evidence for the potential utility of dopamine and L-mimosine in the safer administration of cisplatin.

**\*For correspondence:**
jberman@cheo.on.ca

## Introduction

Despite advances in targeted and personalized cancer therapy, there exists a reliance on classic and relatively broad-spectrum chemotherapies, including platinum agents like cisplatin. Cisplatin is a highly effective anticancer agent that primarily functions through covalent-binding to purine bases, resulting in DNA crosslinks, activation of repair pathways, formation of double stranded breaks (DSBs), increased levels of reactive oxygen species (ROS) and apoptosis (*Dasari and Tchounwou, 2014*). However, cisplatin treatment is associated with a variety of side effects, including dose-limiting toxicities such as oto- and nephrotoxicity (*Brock et al., 2012*; *Pabla and Dong, 2008*). While ototoxicity is permanent in humans, nephrotoxicity can be acute or chronic in nature, and can sometimes demonstrate partial recovery (*Brock et al., 2012*; *Brock et al., 1991*). The former is of particular concern in the treatment of pediatric cancers like neuroblastoma (NBL) or osteosarcoma, when survivors have many developmental life years that may be adversely effected (*Bertolini et al., 2004*; *Brock et al., 2012*; *Finkel et al., 2014*; *Skinner et al., 1998*; *Stöhr et al., 2007*).

Studies suggest that ototoxicity is particularly detrimental in pediatric populations, resulting in a higher frequency of learning disabilities and a decreased self-reported quality of life (*Knight et al., 2005*). Approximately 61% of pediatric patients treated with cisplatin will experience hearing loss

according to the American Speech-Language-Hearing Association (ASHA) criteria (*Knight et al., 2005*). While estimates vary greatly, cisplatin treatment is similarly correlated with changes in glomerular filtration rate (GFR) in approximately 20–30% of patients (*Hartmann et al., 1999*). Hypomagnesemia, or low serum magnesium levels, is a common and persistent side-effect of cisplatin treatment (*Stöhr et al., 2007*), which can lead to seizures, mood changes, tremors, and myocardial infarction (*Lajer and Daugaard, 1999*). Nephrotoxicity was observed in some of the first human trials using cisplatin (*Higby et al., 1974*), and occurs in approximately one third of patients, detectable approximately 10 days after cisplatin therapy (*Crona et al., 2017*; *Gonzales-Vitale et al., 1977*). Unlike ototoxicity, renal damage appears to be at least partially reversible for some patients, usually in those that received low cumulative doses, or were young at the time of treatment (*Brock et al., 1991*; *Latcha et al., 2016*). The ongoing search for oto- and nephroprotective agents highlights the relevance and significance of these toxicities and their mitigation, while maintaining anticancer efficacy (*Kitcher et al., 2019*; *Ou et al., 2010*; *Teitz et al., 2018*; *Thomas et al., 2015*; *Vlasits et al., 2012*).

Currently, there are no FDA-approved drugs to protect against cisplatin ototoxicity. N-acetylcysteine (NAC) is an antioxidant that has shown promise as an oto- and nephroprotective agent in multiple preclinical models (*Dickey et al., 2005*; *Luo et al., 2008*; *Wu et al., 2006*). NAC is currently in a Phase I clinical trial to assess safe dosing for children being treated with cisplatin, with secondary outcome measures of hearing and renal assessment to test for protective effects (NCT02094625). In the context of cisplatin exposure, studies suggest that NAC acts primarily by reducing the elevated levels of ROS that may induce cellular damage, as well as by direct inactivation by complexing with cisplatin in the cell (*Dickey et al., 2005*; *Norman et al., 1992*; *Rushworth and Megson, 2014*; *Sooriyaarachchi et al., 2016*). Sodium thiosulfate (STS) has also demonstrated otoprotective capacity preclinically, and in several pediatric Phase III clinical trials (ex. NCT00716976, NCT00652132) (*Brock et al., 2018*; *Dickey et al., 2005*; *Doolittle et al., 2001*; *Freyer et al., 2017*; *Muldoon et al., 2000*). However, STS is not typically used clinically because of its detrimental impact on overall survival in a subset of patients (*Freyer et al., 2017*). While the reason for this decreased survival is unknown, authors noted that STS may also protect cancer cells from cisplatin-induced toxicity. In attempts to circumvent the potential protection of cancer cells, some studies have attempted to use intra-tympanic administration of protective agents (*Viglietta et al., 2020*).

Options for protecting against nephrotoxicity are also limited. A systematic review of clinical trials suggests that, while hydration is critical for all patients, short-duration, outpatient hydration with potential magnesium supplementation is the safest way to administer cisplatin (*Crona et al., 2017*). Some studies suggest the use of mannitol or furosemide as an osmotic diuretic to prevent renal toxicity, but results are inconsistent and furosemide may itself be ototoxic (*Crona et al., 2017*; *Ding et al., 2016*; *Hayes et al., 1977*; *Santoso et al., 2003*; *Williams et al., 2017*). Regardless of best practice guidelines, renal toxicity remains problematic for patients, especially those with pre-existing kidney dysfunction. One of the main problems in identifying nephroprotective drugs is the lack of a suitable model organism. While murine models have extensive genetic similarities with humans, they are expensive and the kidneys are inaccessible in living animals. On the other hand, human cell lines may be too simplistic to faithfully represent a dynamic organ (*Sharma et al., 2014*; *Yao et al., 2007*).

The zebrafish larva is uniquely suited to identifying novel protective compounds through drug screening, combining the advantages of cell lines (extremely small size suitable for multi-well plates, ease of chemical exposure) and vertebrate models (functional organ systems, genetic conservation) (*MacRae and Peterson, 2015*; *Zon and Peterson, 2005*). Indeed, a zebrafish larval drug screen identified the natural product, visnagin, as a cardioprotectant against short-term anthracycline-induced cardiotoxicty (*Liu et al., 2014*). Further, as a whole-animal drug screening platform, zebrafish can quickly identify compounds with obvious developmental toxicities or absorption issues (*Peterson, 2004*; *Zon and Peterson, 2005*). Zebrafish are particularly well-suited for assessing in vivo oto- and nephrotoxicity as they have analogous organ systems. Zebrafish larvae as young as four days post-fertilization (dpf) possess a functional pronephros structure (a primitive kidney that resembles the two nephrons, running along the length of the larva) and a sophisticated oto-vestibulary system including an inner ear structure with multiple sensory placodes (*Baxendale and Whitfield, 2016*; *Drummond and Davidson, 2010*; *Hentschel et al., 2005*; *Whitfield et al., 2002*). Equally important for in vivo drug screening, like many teleosts, the zebrafish possesses a lateral line

consisting of clusters of hair cells (called neuromasts) for sensing water movement, which are analogous to the hair cells found in mammalian cochlea (*Raible and Kruse, 2000*). Neuromast hair cells are susceptible to ototoxins like cisplatin (*Ou et al., 2007*) and the lateral line is superficial and accessible, making zebrafish an excellent in vivo ototoxicity screening system.

In this study, we report the results of a two-pronged screen incorporating an in vivo ototoxicity assay in zebrafish larvae and an in vitro nephrotoxicity assay using human proximal tubule cells. Over 1200 pharmacologically active compounds were screened to identify candidates that protect against cisplatin-induced nephrotoxicity and ototoxicity. This in vivo ototoxicity screen employed a novel automated Biosorter (similar to a flow cytometer, but used for large particles) to assess larval fluorescence. From the 22 compounds that were both oto- and nephroprotective, we further investigated the utility the endogenous neurotransmitter, dopamine, and the plant amino acid, L-mimosine. Further to their protective effects in vitro and in vivo, we found that these compounds do not impede the ability of cisplatin to induce DSBs and apoptosis in several relevant cancer cell lines, highlighting their potential as putative protective agents that may contribute to the safer administration of cisplatin.

## Results

### Cisplatin treatment results in a dose-dependent decrease in both neuromast health and proximal kidney tubule cell viability

To establish the toxicity profile for cisplatin in both the in vivo ototoxicity screen and the cell-based primary nephrotoxicity screen, initial toxicity curves were generated. In vivo ototoxicity was assessed by examining the health of the lateral line neuromast structures on cisplatin-treated zebrafish larvae, as has been done by others (*Baxendale and Whitfield, 2016*; *Ou et al., 2007*). While other groups have established a dose-response relationship between increased cisplatin dose and decreased larval neuromast health using a semi-quantitative scoring system (*Ou et al., 2010*; *Ou et al., 2007*), the present study utilized the Biosorter to evaluate the fluorescent profiles of experimental larvae to provide an unbiased quantitative measure of neuromast health. For the in vivo screen, 72 hr post-fertilization (hpf) *casper* zebrafish larvae were treated with increasing doses of cisplatin (0–0.05 mM) (*Baxendale and Whitfield, 2016*; *Esterberg et al., 2016*; *Ou et al., 2009*; *Ou et al., 2007*). The following day, 24 hr post-treatment (hpt), larvae were stained with 2 μM YO-PRO-1, and their fluorescence was measured with a Biosorter (*Figure 1a*). A dose-dependent relationship between cisplatin dose and peak height (PH) green fluorescence was observed, which correlated to YO-PRO-1 neuromast staining. The $EC_{50}$, or effective concentration at which half of the maximal neuromast PH fluorescence was calculated to be 0.027 mM, according to a four-parameter log-logistic model (see Materials and methods for supporting information). Data from the same experiment completed 48 hpt showed a similar dose–response relationship and can be found in *Figure 1—figure supplement 1a*.

Similarly, HK-2 human proximal tubule cells were treated with increasing concentrations of cisplatin (0–0.015 mM). An alamarBlue assay to determine cell viability was undertaken at 48 hpt (*Figure 1b*). Results displayed a dose-dependent decrease in HK-2 cell viability with increasing cisplatin dose. Data from the same experiment completed 24 hpt demonstrated a similar dose–response relationship, and can be found in *Figure 1—figure supplement 1b*.

### Two-pronged drug screen reveals 22 compounds that are both oto- and nephroprotective

Following the establishment of predictable dose–response curves for cisplatin-induced oto- and nephrotoxicity, the Sigma LOPAC[1280] library was screened for potential protective agents. This was accomplished with a parallel two-pronged approach using our assays described above in zebrafish neuromasts and HK-2 human proximal tubule cells.

For the in vivo ototoxicity study, four larvae per well were treated with either the vehicle control or one of the compounds from the Sigma LOPAC[1280] library at a final concentration of 0.01 mM. Three hours later, larvae were then treated with vehicle control, or cisplatin (0.02 mM, the approximate $EC_{50}$ in the YO-PRO1 assay, where neuromast fluorescence was approximately 0.5 fold of control values), such that larvae were either treated with cisplatin alone, or cisplatin + the library

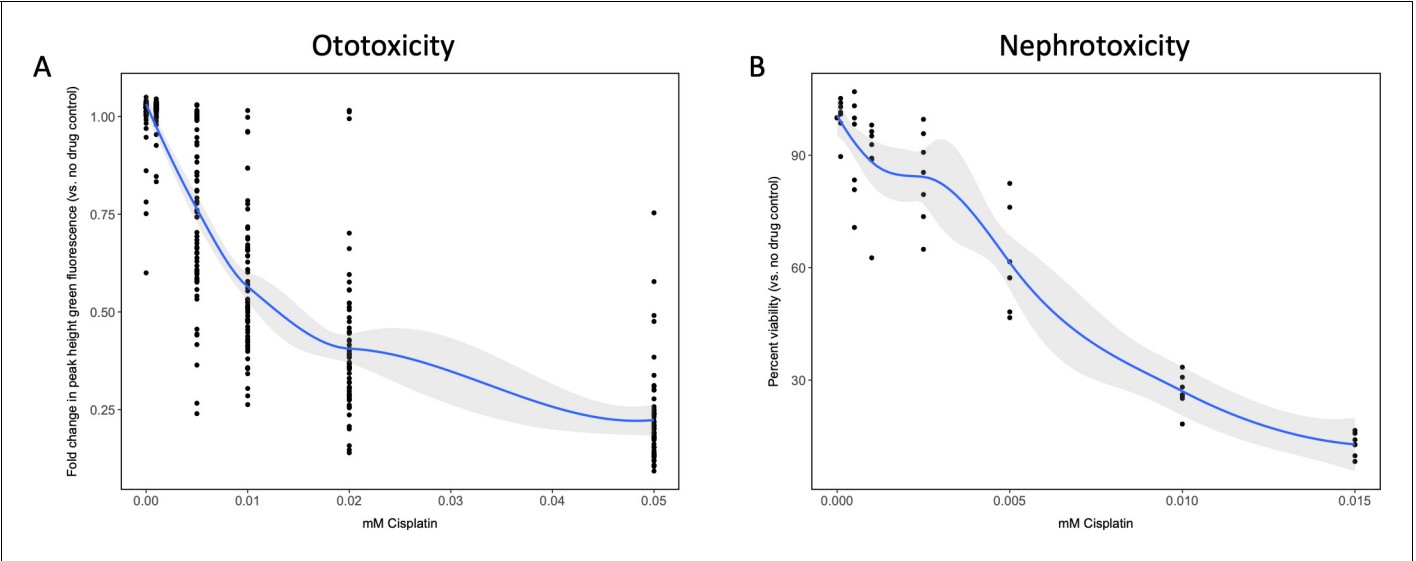

**Figure 1.** Dose–response curves demonstrate decreasing neuromast integrity and human proximal tubule cell viability with increasing doses of cisplatin. (A) Groups of approximately 50 *casper* zebrafish larvae were treated with increasing doses of cisplatin, by addition to the E3 media surrounding the larvae, at 72 hr post-fertilization (hpf). The following day, larval neuromasts were stained with 2 µM YO-PRO1, then were subjected to Biosorter-mediated fluorescence profiling. Peak Height (PH) of green fluorescence is displayed, relative to untreated controls. Each data point represents an individual larva. Dose–response relationship is represented by the blue line, which was calculated with a four-parameter log-logistic model, as described in a relevant study (**Ritz et al., 2015**). Modeling was done in R with a *drc* extension package. Grey-shaded area represents the 95% confidence interval (CI) of this line. (B) HK-2 human proximal tubule cells were treated with increasing concentrations of cisplatin for 48 hr. Cells were rinsed, then an alamarBlue assay was performed as per the manufacturer's instructions. Data are represented as % viability, in comparison with untreated cells. N = 4, an average of at least two wells was measured per replicate. Dose–response analysis performed as in A). The online version of this article includes the following figure supplement(s) for figure 1:

**Figure supplement 1.** Dose–response curves demonstrate decreasing neuromast integrity and human proximal tubule cell viability with increasing doses of cisplatin.

compounds. At the experimental endpoint, larvae were stained with YO-PRO1, then biosorted to measure PH green fluorescence. The average fold change in neuromast fluorescence, compared to the no drug control value, of the four larvae treated with each compound can be seen in *Figure 2a*. Toxic compounds (defined as resulting in death of at least ¾ larvae, when administered with 0.02 mM cisplatin) can be found listed in *Supplementary file 1*.

Human proximal tubule HK-2 cells were either untreated, treated with the approximate $EC_{50}$ of cisplatin (0.005 mM) alone, or both the $EC_{50}$ of cisplatin + each of the compounds from the drug library, at a final concentration of 0.01 mM (*Figure 2b*).

Following completion of both screens, we identified 22 'hits': compounds that were protective against cisplatin nephrotoxicity and ototoxicity (resulting in viability values corresponding to 0.9–1.1X of the no drug control value). For these 22 compounds, their main biological mechanism of action (as defined by the Sigma LOPAC$^{1280}$ library), and their current clinical usage (if applicable) can be found in *Table 1*. Dopamine and L-mimosine were revealed to be two of the most protective compounds, based on the results from the in vitro and in vivo drug screen. These are both members of the dopaminergic pathway and share a similar chemical structure (*Figure 2c*). Interestingly, members of the dopaminergic pathway were frequently identified in the list of protective compounds, including dopamine, GBR-12909 dihydrochloride (Vanoxerine, a dopamine reuptake inhibitor), p-Fluoro-L-phenylalanine (substrate for tyrosine hydroxylase), and L-mimosine (inhibitor of dopamine β-hydroxylase [*Hashiguch and Takahashi, 1976*]). Treatment with any of these compounds may result in increased dopamine signaling. An overview of the potential interaction between these compounds can be seen in *Figure 2*.

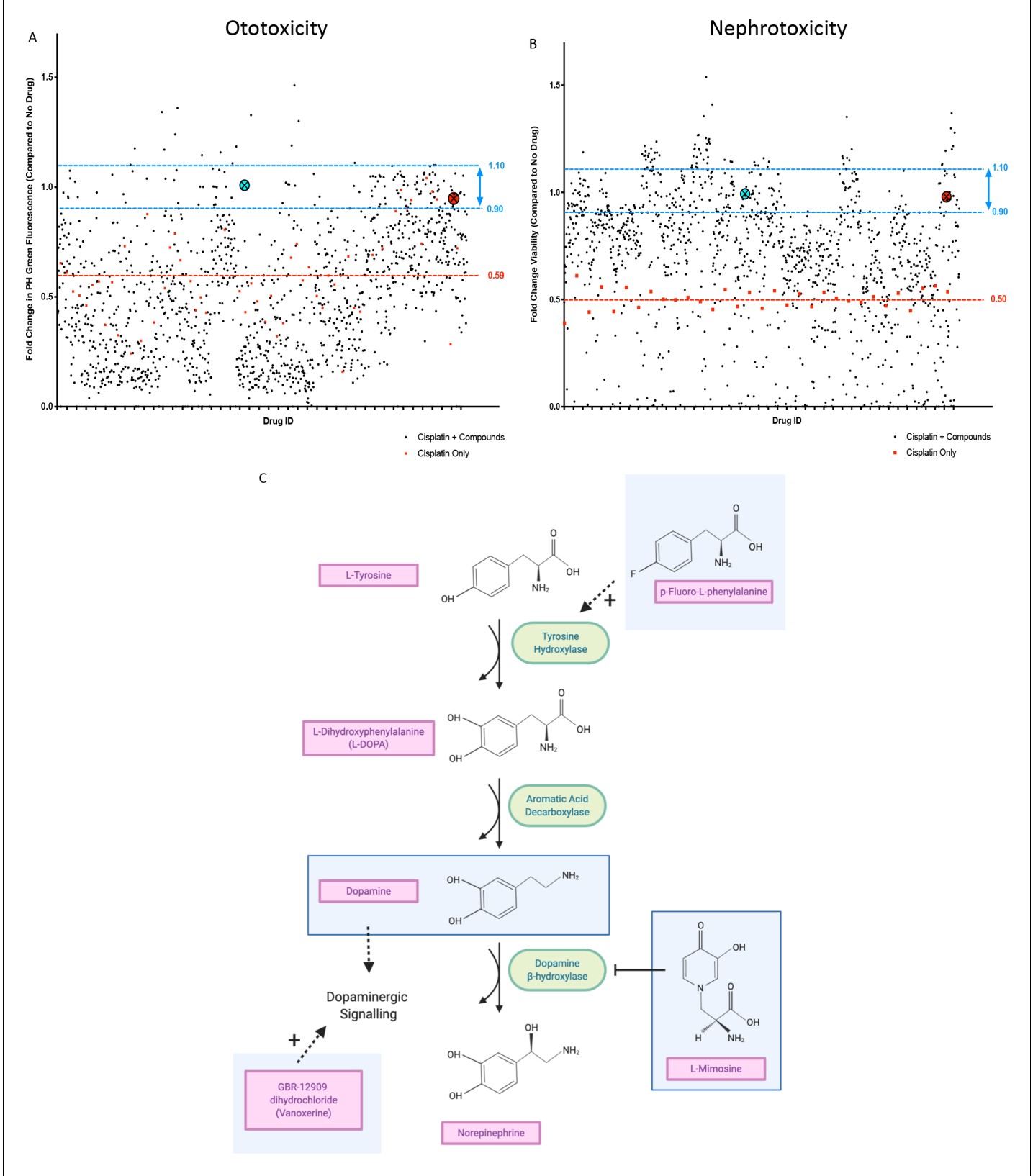

**Figure 2.** In vivo ototoxicity and in vitro nephrotoxicity drug screens reveal 22 compounds that are potentially oto- and nephroprotective, including dopamine and L-mimosine. (**A**) Zebrafish larvae were pretreated with either vehicle control, or each of the compounds from the Sigma LOPAC1280 library of pharmacologically active compounds at a final concentration of 0.01 mM. Three hours later, at 72 hr post-fertilization (hpf), larvae were treated

*Figure 2 continued on next page*

Figure 2 continued

with either vehicle control or cisplatin, at a concentration of 0.02 mM. 48 hr later, larval neuromasts were stained with YO-PRO1 and subjected to fluorescence profiling using a Biosorter. Peak Height (PH) green fluorescence is displayed, in comparison with untreated larvae. Each compound was tested on four larvae and the average is displayed. The aqua point in the X represents dopamine hydrochloride and the red point with the X represents L-mimosine. The blue lines correspond to 0.9–1.1 fold of the control value and the red line corresponds to the average of fish treated with cisplatin alone. (B) HK-2 kidney proximal tubule cells were either treated with vehicle control, 0.005 mM cisplatin alone, or 0.005 mM cisplatin + each of the compounds from the Sigma LOPAC1280 drug library at a final concentration of 0.01 mM. Two days later, an alamar Blue assay was performed according to the manufacturer's instructions to determine cell viability. Fold change in viability (in comparison with vehicle control treated cells) is displayed. The average of two wells was used per drug. Highlighted points and lines correspond to those in A). (C) The dopamine biosynthesis pathway consists of intermediate molecules and enzymes (indicated with green ovals). The compounds that were hits in both assays are shown in light blue boxes. L-mimosine is able to inhibit dopamine beta-hydroxylase. P-fluoro-L-phenylalanine can act as a substrate for tyrosine hydroxylase. GBR-12909 dihydrochloride (aka. Vanoxerine) is a selective dopamine reuptake inhibitor. All these compounds could have the net pharmacological effect of increasing available dopamine levels.

## Follow-up assessment highlights the otoprotective effects of dopamine and L-mimosine

To examine the effects of dopamine and L-mimosine, we completed pilot experiments on both the ideal treatment schedule and optimal doses for these protective agents in vivo (data not shown). Based on these preliminary results, we determined that a 12 hr pretreatment with the protective agent prior to cisplatin treatment was ideal, so all subsequent experiments used this time course. Following this schedule, we observed significant neuromast protection with dopamine at 0.03 mM, L-mimosine at both 0.03 and 0.04 mM, and with the combination of dopamine and L-mimosine at 0.02 mM each (*Figure 3a*), when compared to larvae treated with cisplatin alone. Representative images of these larvae at the indicated treatment concentrations can be found in *Figure 3b*. Since none of the treatment doses resulted in complete protection of the neuromasts, in attempts to determine whether these results were biologically meaningful, we assessed the neuromast integrity following treatment with 0.01 mM cisplatin, or 50% of the dose administered to the larvae receiving the protective compounds (*Figure 3a*, column 1). Treatment with the full dose of cisplatin (100%, or 0.02 mM) and either L-mimosine at 0.04 mM or the combination of dopamine and L-mimosine at 0.02 mM each resulted in neuromast fluorescence that was not significantly different from the larvae treated with 0.01 mM cisplatin alone (the 50% dose). These data suggest that use of these protective agents could allow for larger doses of cisplatin to be given without further damage to the hair cells, which could have potential therapeutic benefit.

To assess if the same protection conferred on the lateral line neuromast structures in zebrafish larvae was preserved in the zebrafish inner ear hair cells, treated larvae were stained with Alexa Fluor 488 phalloidin to visualize filamentous actin. Briefly, *casper* larvae were pretreated at 60 hpf with either vehicle control, or 0.03 mM of dopamine or L-mimosine, then treated at 72 hpf with 0.01 mM cisplatin for 48 hr, then fixed and stained and imaged with confocal microscopy (*Figure 4a-d*). Qualitatively, the hair cells appear more plentiful and more structurally intact in the groups that were pretreated with dopamine and L-mimosine in comparison to the cisplatin alone group.

We were subsequently able to quantify the difference in hair cell morphology using Imaris v.X64 9.1.2 software for surface reconstruction. Ellipticity (prolate, or elongation around the long axis) was used as a measure of the status of the hair cells. Intact and healthier hair cells exhibit a higher prolate ellipticity, or vertical ellipticity. Results demonstrate that cisplatin alone decreases the prolate ellipticity in comparison with control hair cells (*Figure 4e*). Pretreatment with either dopamine or L-mimosine significantly increased the prolate ellipticity, in comparison with cisplatin treatment alone. Importantly, hair cells from larvae that were pretreated with dopamine or L-mimosine had prolate ellipticity that was not significantly different from hair cells of control larvae (*Figure 4e*). In order to ensure that the ellipticity measurements reflected the same trends that we saw with overall hair cell number, we used the Cell Counter plugin for ImageJ to quantify the number of hair cells in the same maximum-projection z-stack images that were used for Imaris analysis (*Figure 4f*). It should be noted that the same trend was observed, where dopamine and L-mimosine pretreatments resulted in hair cell numbers that were higher than those observed in larvae treated with cisplatin alone.

**Table 1.** Compounds from the Sigma LOPAC[1280] compound library that were hits in both the oto- and nephroprotection assays. Clinical usage information obtained from the ChEMBL database (*Gaulton et al., 2017*).

| Drug ID | Drug name | Biological action | Clinical usage | Notes |
|---|---|---|---|---|
| 1, D06 | 5-Azacytidine | DNA methyltransferase inhibitor | Approved for use in myelodysplatic syndrome (MDS), chronic myelomonocytic leukemia (CML), some advanced solid tumours | |
| 1, E02 | (±)-Nipecotic acid | GABA uptake inhibitor | N/A | |
| 1, F07 | 1-Aminobenzotriazole | Cytochrome P450 and chloroperoxidase inhibitor | N/A | Potent cytochrome P450 inhibitor, used in research (*Ortiz de Montellano, 2018*) |
| 1, G06 | Apigenin | Arrests cell cycle at G2/M phase | N/A | Protects against cisplatin-induced nephrotoxicity in preclinical models (*Hassan et al., 2017*; *Ju et al., 2015*) and may have anticancer activities summarized by *Yan et al., 2017* |
| 3, F04 | Betamethasone | SAID (steroidal anti-inflammatory drug); glucocorticoid | Approved for use in obstructive lung disease, nasal obstruction, eye diseases, eczema and psoriasis | Historically showed some efficacy as an anti-emetic agent in chemotherapy regimens (*Sorbe, 1988*) |
| 5, F10 | Ganaxolone | Positive allosteric modulator of GABA-A receptors | N/A | Reached Phase III clinical trials for use in drug resistant partial onset seizures (NCT01963208) |
| 6, E07 | GBR-12909 dihydrochloride (Vanoxerine) | Selective dopamine reuptake inhibitor | N/A | Investigated as a potential treatment for cocaine-abuse disorder (NCT00218049) |
| 7, B03 | N-Ethylmaleimide | Sulfhydryl alkylating agent that inactivates NADP-dependent isocitrate dehydrogenase and many endonucleases | N/A | |
| 7, E04 | Phenserine | Selective, non-competitive acetylcholinesterase (AChE) inhibitor | N/A | |
| 7, G06 | p-Fluoro-L-phenylalanine | Substrate for tyrosine hydroxylase; arrests cells at G2 | N/A | |
| 8, C07 | Dopamine hydrochloride | Endogenous neurotransmitter | Approved for use in shock caused by heart attack, trauma, surgery, heart failure, and kidney failure | Historically investigated as a potential nephroprotective agent for use with cisplatin (*Baldwin et al., 1994*; *Somlo et al., 1995*), may influence neurotransmission at sensory hair cells (*Toro et al., 2015*) and is known to be involved in auditory processes (*Gittelman et al., 2013*) |
| 9, H05 | SB-525334 | A potent activin receptor-like kinase (ALK5)/type I TGFß-receptor kinase inhibitor | N/A | |
| 12, G04 | Ouabain | Blocks movement of the H5 and H6 transmembrane domains of $Na^+$-$K^+$ ATPases | N/A | Historical use for myocardial infarction and angina treatment, used in the treatment of digitalis intoxication (*Fürstenwerth, 2010*), used in poison darts in eastern Africa, defined as an 'extremely hazardous substance' in the USA |

*Table 1 continued on next page*

*Table 1 continued*

| Drug ID | Drug name | Biological action | Clinical usage | Notes |
|---|---|---|---|---|
| 14, E04 | Ritanserin | Potent 5-HT2 serotonin receptor antagonist which passes the blood-brain barrier | N/A | Investigated for use in cocaine dependence (NCT00000187) |
| 14, E08 | CP-335963 | Aurora two kinase inhibitor, PDGF inhibitor, and anti-proliferative | N/A | |
| 15, C11 | Tyrphostin 1 | EGFR tyrosine kinase inhibitor | N/A | Tyrphostins have been shown to reduce small intestinal damage by cisplatin and 5-Fluorouracil (5-FU) (*Zlotnik et al., 2005*) |
| 15, G02 | '1-(1-Naphthyl)piperazine hydrochloride | 5-HT2 serotonin receptor antagonist | N/A | |
| 16, D03 | Trifluoperazine dihydrochloride | Xanthogenate derivative with in vivo anti-tumor and anti-HIV activity; inhibits phospholipase D and phosphatidylcholine phospholipase C (PIPLC) | Approved for use as an antipsychotic for individuals with schizophrenia, some use as an anxiolytic, $D_2$ receptor antagonist | Has anti-adrenergic, anti-dopaminergic and anti-cholinergic effects. Thought to minimize hallucinations and delusions through inhibition of the $D_2$ receptors in the mesocortical and mesolimbic pathways |
| 16, D07 | S(-)-UH-301 hydrochloride | Potent and selective 5-HT1A serotonin receptor antagonist | N/A | |
| 16, D09 | L-Mimosine from Koa hoale seeds | Potential inhibitor of the cell cycle giving rise to growth arrest in G1-phase. An iron chelator that inhibits DNA replication in mammalian cells. Has been shown to have apoptotic activity in xenotransplanted human pancreatic cancer | N/A | Inhibits copper containing enzymes tyrosinase and dopamine β-hydroxylase (*Hashiguch and Takahashi, 1976*), shown to re-activate hypoxia-inducible factor 1α (HIF-1α) and reduce renal fibrosis in a rat model of renal ablation (*Yu et al., 2012*), blocks proliferation in prostate carcinoma cells (*Chung et al., 2012*) and breast cancer cells (*Kulp and Vulliet, 1996*) |
| 16, D10 | AC-55649 | Subtype selective retinoic acid receptor beta2 (RARbeta2) agonist | N/A | RARβ may have tumour suppressor activities (*Alvarez et al., 2007*) |
| 16, F11 | Caroverine hydrochloride | Nonselective NMDA and AMPA glutamate receptor antagonist | Used to treat muscle spasms and tinnitus in some countries, not FDA approved | Has antioxidant properties (*Udilova et al., 2003*), attenuates noise-induced hearing loss in rats (*Duan et al., 2006*), treats tinnitus in humans (*Denk et al., 1997*) |

## Follow-up examination of in vivo glomerular filtration supports the nephroprotective effects of dopamine and L-mimosine

Our group has adapted a FITC-tagged inulin-based glomerular filtration rate (GFR) assay (*Hentschel et al., 2005*) for medium-high throughput use in zebrafish larvae. Inulin is a polysaccharide that is excreted exclusively through the nephron, so when this compound is injected into the circulation, decreases in vascular fluorescence over time correspond with GFR (*Hentschel et al., 2005*; *Rider et al., 2012*). Previous studies have used ImageJ analysis to quantify fluorescence in individual larvae over time as a measure of GFR (*Hentschel et al., 2005*; *Rider et al., 2012*), which is a time-consuming method that restricts the number of possible experimental animals. To first reproduce these findings, 72 hpf *casper* zebrafish were treated with either vehicle control or 0.125 mM cisplatin for 24 hr, injected with FITC-inulin via the common cardinal vein, then were imaged immediately following injection and at 2 hr post-injection. Cisplatin treatment resulted in a decreased change in fluorescence over time, indicative of decreased GFR (*Figure 5—figure supplement 1a*). Next, to increase the throughput of this assay, we performed the same experiment using the Biosorter for fluorescence measurements, where we observed the same trend (*Figure 5—figure supplement 1b*). Since this GFR assay is fundamentally linked with cardiovascular function and heart rate, heart rate

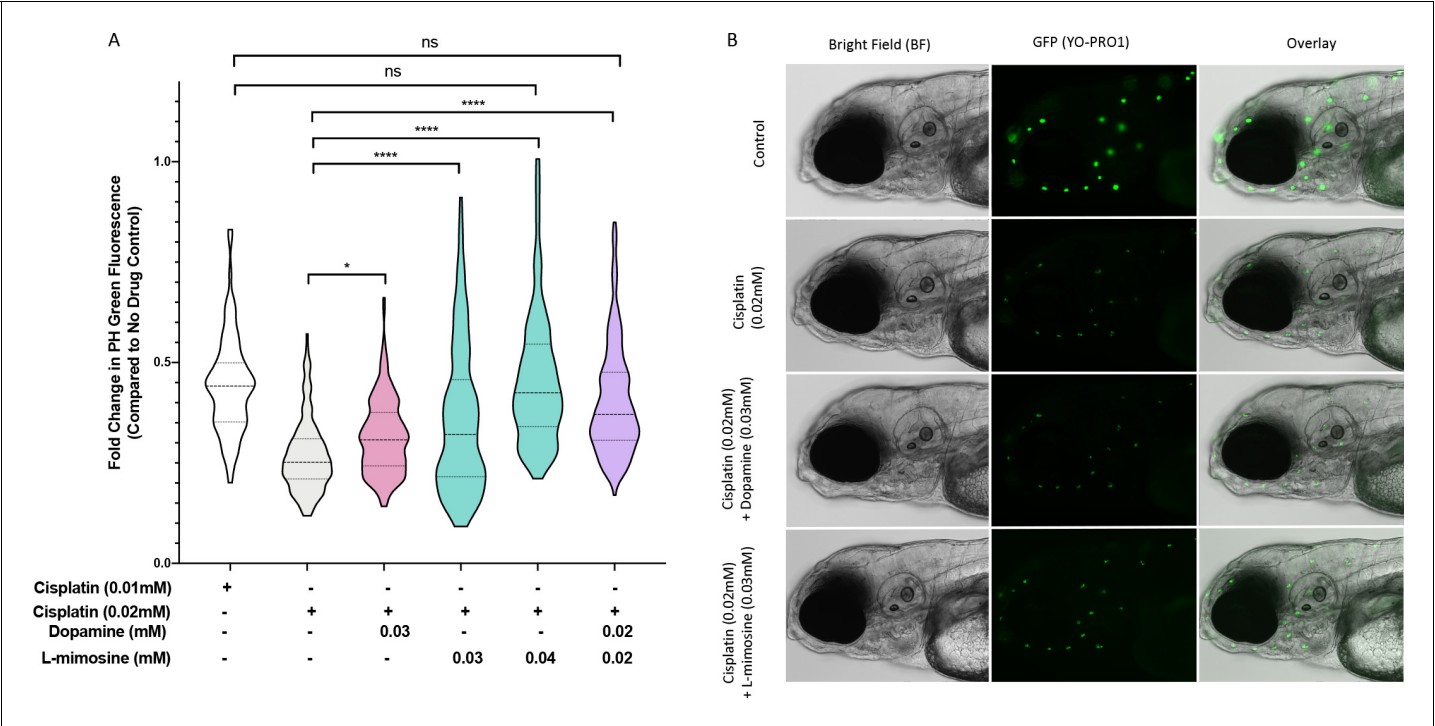

**Figure 3.** Dopamine and L-mimosine pretreatment partially protects zebrafish larval neuromasts from cisplatin-induced damage. (**A**) Following dose and scheduling optimization, groups of larvae at 60 hr post-fertilization (hpf) were treated with either vehicle control or the indicated concentrations of protective agents. Larvae were rinsed then treated with either vehicle control or cisplatin at the indicated concentration at 72 hpf. The following day, the larval neuromasts were stained with 2 μM YO-PRO1 then subjected to fluorescence profiling using a Biosorter. Fold change in Peak Height (PH) fluorescence is displayed, compared to untreated controls.*=p<0.05, ****=p<0.001, as per Kruskal-Wallis with a Dunn's multiple comparison test between indicated groups. Violin plot displays the median in a solid line and the interquartile range with hashed lines, with surrounding data points outlined by the shape. N=3, average of 75 larvae/treatment/replicate. (**B**) Representative images of larvae measured in A), with the treatment types as shown, viewed with brightfield (BF) or fluorescence (green neuromasts, YO-PRO1), or overlay of BF and fluorescence. Images acquired with an Axio Observer Z1 microscope at 20X.

was also measured in each of the treatment groups, and there were no detectable differences between treatments (*Figure 5—figure supplement 1c*). To measure the potential protective effects of dopamine and L-mimosine, larvae were pretreated with 0.03 mM of either protective compound at 60 hpf, then treated with 0.125 mM cisplatin at 72 hpf. Following inulin injection, there was a significant decrease in fluorescence in the control group over 2 hr demonstrating normal excretion, but not in the group that was treated for 24 hr with 0.125 mM cisplatin (*Figure 5a*). However, when the larvae were pretreated with either dopamine or L-mimosine prior to cisplatin treatment, there was a detectable decrease in fluorescence over 2 hr, demonstrating enhanced GFR (*Figure 5*, representative images of inulin-injected larvae can be found in *Figure 5—figure supplement 1d*.)

To determine whether the alterations in GFR were associated with histological changes in the pronephros structure, coronal sections of fixed, treated larvae stained with H and E were imaged. Results suggested that both 24 and 72 hr treatment with 0.125 mM cisplatin did not cause any distinct visible changes to the pronephros histology (24 hpt data: *Figures 5b–e*, 72 hpt data: *Figure 5—figure supplement 2a–d*).

## Dopamine and L-mimosine do not inhibit cisplatin-induced cancer cell cytotoxicity

When establishing the utility of chemoprotective agents, it is imperative to ensure that the adjuvant drugs do not also guard the cancer cells from the anticancer effects of the compound in question. To assess this, alamarBlue viability assays were performed in several cancer cell lines representative of cancers that are commonly treated with cisplatin, including NBL cell lines (SK-N-AS and LAN5) and an oral squamous cell carcinoma cancer cell line (HSC-3s) with various doses of cisplatin +/-

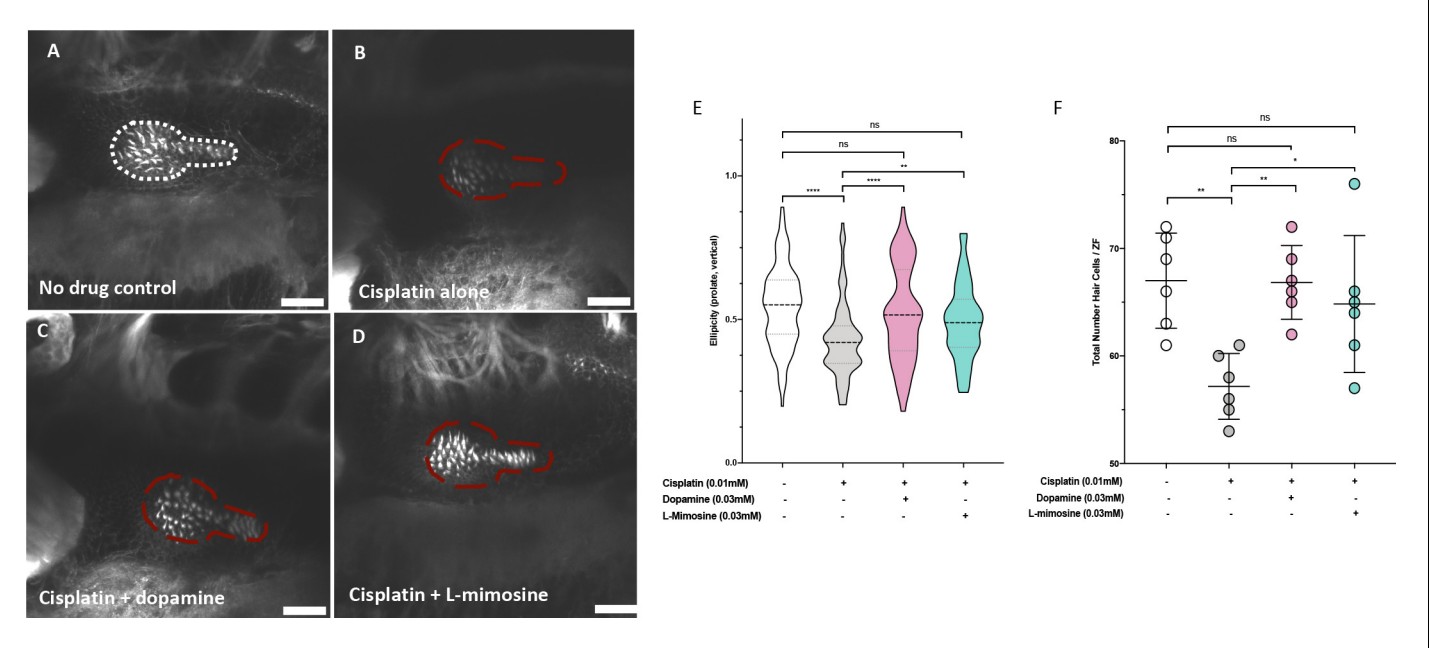

**Figure 4.** Dopamine and L-mimosine pretreatment protects zebrafish inner ear hair cells from cisplatin-induced damage. *Casper* zebrafish larvae were first pretreated at 60 hr post-fertilization (hpf) with either vehicle control (**A and B**), 0.03 mM dopamine (**C**), or 0.03 mM L-mimosine (**D**). Larvae were then rinsed at 72 hpf and treated with either vehicle control (**A**) or 0.01 mM cisplatin B-D). Two days later (48 hr cisplatin treatment), larvae were fixed, permeabilized, then stained with AlexaFluor488-Phalloidin, then imaged with a Zeiss LSM510 confocal microscope. Larvae mounted laterally with ventral side at the top. Scale bar = 20 μM. N=6/treatment. White line dashed represents the average posterior macula placode of three control larvae, red dashed line represents this average area superimposed onto other treatments. Note that these images were taken with the exact same settings to ensure accurate representation. (**E**) To perform topological analysis of the hair cells, the original Zeiss LSM files of 2D images ranging from 27 to 42 μm in depth were imported to Imaris v.X64 9.1.2 software for surface reconstruction. Ellipticity (prolate, or elongation around the long axis) was used as a measure of the status of the hair cells with healthier hair cells exhibiting a higher prolate ellipticity. \*\*=p<0.01, \*\*\*\*=p<0.001, as per Kruskal-Wallis testing with a Dunn's multiple comparison test. (**F**) Hair cell numbers were counted using the Cell Counter plugin in ImageJ using the same maximum projection images used in E). Individual points represent the number of hair cells/larvae. \*=p<0.05, \*\*=p<0.01, and \*\*\*\*=p<0.001, as per one-way ANOVA with a Tukey post-test. N = 6 larvae/treatment with 16 hair cells measured/larvae.

dopamine or L-mimosine. In all treatment groups, and at both 24 and 48 hpt, dopamine and L-mimosine either had no effect on cancer cell viability, or significantly increased the cytotoxic effects of cisplatin (48 hpt data: *Figures 6a–f*, 24 hpt data: *Figure 6—figure supplement 1a–f*).

Cisplatin is known to induce cancer cell death through apoptosis. Apoptosis was measured using Annexin-V labeling and flow cytometry. We found increased levels of apoptosis in NBL cells treated with cisplatin alone (*Figure 6g*), and pretreatment with either 0.03 mM dopamine or L-mimosine did not reduce the levels of apoptosis.

Cisplatin is a DNA-binding agent that causes inter- and intra-strand crosslinks in cellular DNA. The repair of inter-strand crosslinks typically involves the formation of at least a temporary DSB, which result in the phosphorylation of histone protein H2AX, called γH2AX (*Rogakou et al., 1999*; *Rogakou et al., 1998*). Levels of γH2AX antibody staining 24 hr following cisplatin exposure has been reported to correlate with the long-term viability of the cells following treatment (*Olive and Banáth, 2009*). To assess the contribution of DSB formation to cisplatin-induced apoptosis, we used an γH2AX-based immunofluorescence assay. Results in both SK-N-AS and LAN5 NBL cell lines demonstrate that cisplatin treatment increased DSB formation, and pretreatment with either dopamine or L-mimosine did not reduce these levels (Quantification of DSBs: *Figure 6h,i*, Representative images of SK-N-AS cells: *Figure 6j*). However, there were differences between the cell lines based on their response to L-mimosine pretreatment: in SK-N-AS cells, L-mimosine pretreatment increased DSB formation over cisplatin treatment alone, while in LAN5 cells, the same treatment resulted in a slight, but significant decrease in DSB formation compared with cisplatin alone.

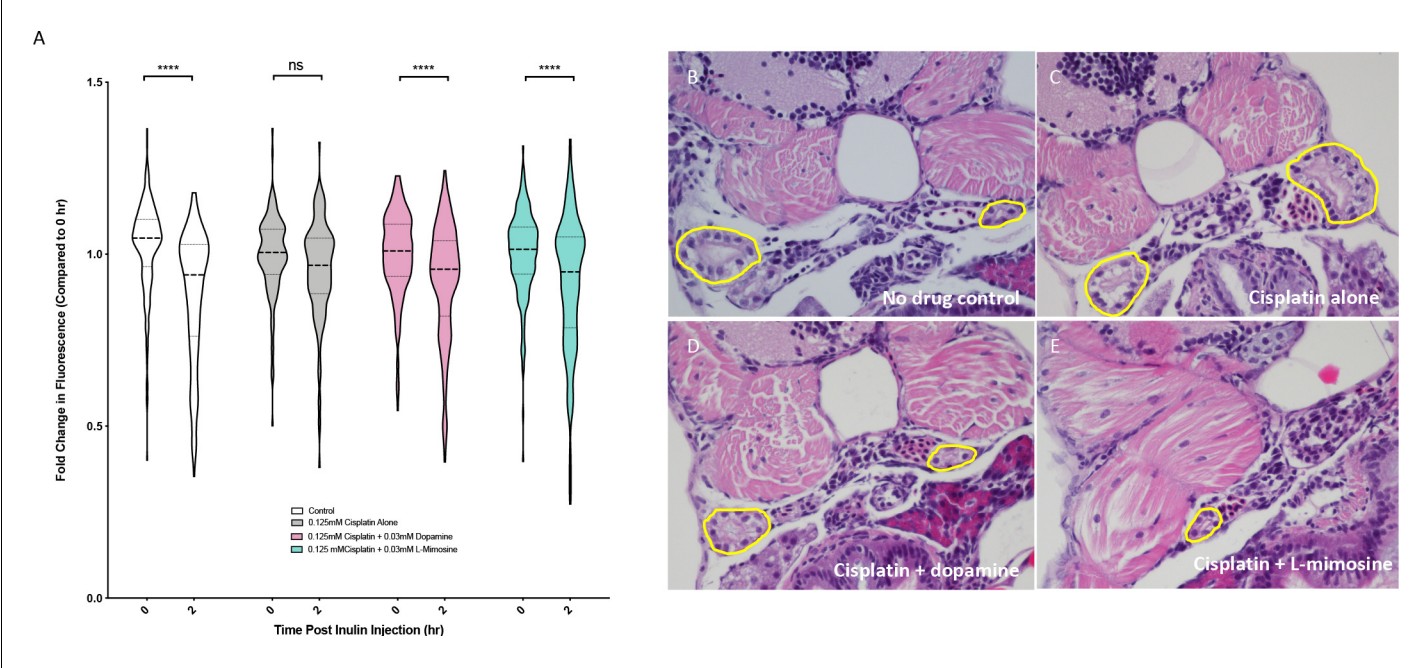

**Figure 5.** Dopamine and L-mimosine pretreatment preserves the glomerular filtration rate (GFR) from cisplatin-induced damage, but cisplatin treatment has no detectable change on pronephros histology. (**A**) *Casper* zebrafish larvae were treated at 60 hr post-fertilization (hpf) with either vehicle control or the indicated protective agents. At 72 hpf, larvae were rinsed then treated with either vehicle control or 0.125 mM cisplatin for 24 hr. Larvae were then injected via the common cardinal vein with FITC-inulin, then measured for fluorescence swiftly with the Biosorter. Larvae were rinsed then measured 2 hr later. Fold change in overall larval fluorescence is represented in relation to 0 hr. ****=p<0.001, as per two-way ANOVA with a Tukey post-test. Three replicates, with 50 larvae/treatment group/time point minimum. Representative images of larvae can be found in *Figure 5—figure supplement 1d*. Larvae were treated as in (**A**) and were fixed at either 24 hr post-treatment (**B–E**) or at 72hpt (*Figure 5—figure supplement 2a-d*). Larvae were pre-embedded in low melting point agarose, then in paraffin, then sectioned and stained with H and E. (**B**) Control, (**C**) Cisplatin only, (**D**) Cisplatin + 0.03 mM dopamine, (**E**) Cisplatin + 0.03 mM L-mimosine. No significant differences were observed in the proximal tubular histology. The online version of this article includes the following figure supplement(s) for figure 5:

**Figure supplement 1.** Optimization of the experimental detection of cisplatin-induced decreases in glomerular filtration rate (GFR) in zebrafish larvae.

**Figure supplement 2.** Zebrafish pronephros histology does not look significantly different following treatment with cisplatin or either protective agent at 3 days post-treatment (dpt).

## Discussion

While personalized pharmacology and targeted cancer treatments are becoming a more realistic possibility for cancer therapy, nonspecific cytotoxic chemotherapy agents like cisplatin are unlikely to be completely replaced in the foreseeable future. Cisplatin is used in the treatment of multiple cancer types, including several pediatric malignancies like NBL, osteosarcoma and in germ cell tumors (*Kelland, 2007*). However, like most cytotoxic drugs, cisplatin has a host of toxic side effects that are sometimes considered to be 'collateral damage.' Nephrotoxicity and ototoxicity are among the common dose-limiting side effects of cisplatin treatment. Approximately 30% of patients experience some type of cisplatin-induced nephrotoxicity, while ototoxicity occurs in at least 60% of pediatric patients (*Hartmann et al., 1999*; *Knight et al., 2005*). In order to continue harnessing the utility of cisplatin, there have been many efforts to identify protective compounds.

STS is an otoprotective agent that was recently tested in a pediatric clinical trial (*Freyer et al., 2017*). While the individuals that received STS had significantly less hearing loss (45 vs. 84%, p=0.00022), the individuals with disseminated disease at the time of diagnosis had worse overall survival when treated with STS (45% vs. 84%, respectively, p=0.009). It is not currently known why this difference in overall survival with STS treatment occurred, but it is worth noting that STS is known to function as a protective agent in several ways, both scavenging ROS and forming a direct complex with cisplatin, inactivating the drug (*Bijarnia et al., 2015*; *Sooriyaarachchi et al., 2012*). While the direct complexing with cisplatin was known at the outset of the clinical trial, results from preclinical

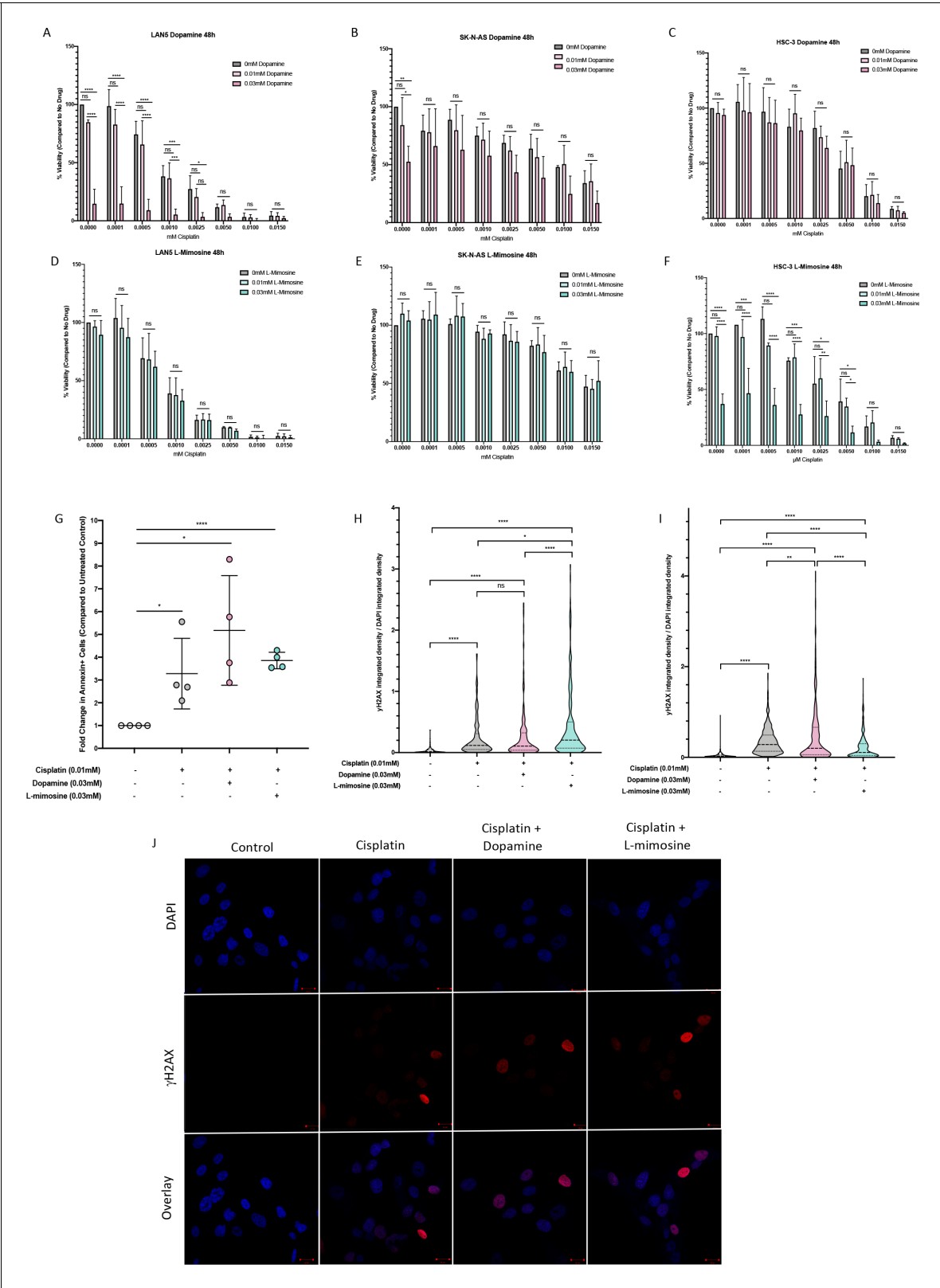

**Figure 6.** Dopamine and L-mimosine do not protect neuroblastoma (NBL) and oral squamous cell carcinoma cell lines from cisplatin-induced cytotoxicity. NBL and oral squamous cell carcinoma cell lines were pretreated for 12 hr with vehicle control, or either dopamine or L-mimosine at either 0.01 mM or 0.03 mM. Cells were then treated with increasing concentrations of cisplatin and incubated for either 24 hr (*Figure 6—figure supplement 1a-f*) or 48 hr (data shown here). (**A–F**) An alamarBlue assay was used to determine cell viability. Results are displayed as % compared to untreated

*Figure 6 continued on next page*

*Figure 6 continued*

control. (**A and D**) LAN5, (**B and E**) SK-N-AS, (**C and F**) HSC-3. *=p<0.05, **=p<0.01, ***=p<0.005, and ****=p<0.001, as per two-way ANOVA with a Tukey post-test. N=3. (**G**) SK-N-AS cells were treated the same as in A-F, with the concentrations of protective agent specified. 48 hr following cisplatin treatment, cells were prepared for PE-conjugated Annexin V/SYTOXBlue-based flow cytometry. Graph of fold change in Annexin+ cells (gated to biological control) is displayed, relative to vehicle control. *=p<0.05, ****=p<0.001, as per two-tailed student's t-tests comparing treatment groups to control. N=4. Representative flow plots and gating strategies can be found in *Figure 6—figure supplement 2a-i* **H–J**) Cancer cells were treated as in A-G, with the concentrations of protective agents specified. Twenty four hours following cisplatin treatment, cells were fixed, permeabilized and labeled with anti-phospho-histone H2A.X (Ser139) and DAPI to label γH2AX positive foci and nuclear material, respectively. (**H and I**) Quantification of γH2AX staining, reported as γH2AX integrated density/DAPI integrated density, with each data point corresponding to an individual nucleus. *=p<0.05, **=p<0.01, ****=p<0.001, as per Kruskal-Wallis testing with a Dunn's multiple comparison test. N=3. (**H**) SK-N-AS cells, (**I**) LAN5 cells. (**J**) Representative confocal microscopy of SK-N-AS cells with indicated treatments, displaying DAPI, gamma H2AX, and an overlay.

The online version of this article includes the following figure supplement(s) for figure 6:

**Figure supplement 1.** Dopamine and L-mimosine do not protect cancer cell lines from cisplatin-induced death 24 hr following cisplatin treatment.
**Figure supplement 2.** Representative flow plots and gating strategies for the detection of late-stage apoptosis and cell death in SK-N-AS neuroblastoma (NBL) cells.

trials suggested that delayed administration of STS (4–8 hr following cisplatin therapy) would prevent any unwanted cancer cell protection (*Dickey et al., 2005*; *Freyer et al., 2017*; *Muldoon et al., 2000*). This highlights the delicate nature of identifying a protective agent to be given with a potentially life-saving chemotherapy – the adjuvant therapy must not interfere with the original compound. Thus, it would be prudent to identify compounds with a novel mechanism of oto- and/or nephroprotection, which may be best accomplished using an unbiased screening approach, rather than rational or target-based drug design.

Our study used a novel two-pronged drug screen approach that employed both a prototypical cell-based nephrotoxicity assay and a novel Biosorter-based fluorescence in vivo ototoxicity assay. Human studies suggest that over the initial 24 hr period following administration, plasma cisplatin levels range from approximately 1–10 μg/mL, which translates to 0.0033 mM-0.033 mM (*Lanvers-Kaminsky et al., 2006*; *Rajkumar et al., 2016*), well within the range of what we used experimentally. By using both human cells and animal models, we were able to focus the list of potentially protective compounds to those that were likely to have good absorption and low levels of systemic toxicity at effective doses. For example, there were seven compounds in the Sigma LOPAC[1280] library that were hits in the in vitro nephrotoxicity assay that were toxic at the same dose to the zebrafish larvae (listed in *Supplementary file 1*). It should also be noted that since these compounds were not assessed without cisplatin, we can only conclude that these compounds were toxic to approximately 75% of larvae when administered at a concentration of 0.01 mM in combination with 0.02 mM cisplatin. While this does not indicate that these compounds are not useful, it does suggest that their toxicity profiles may be challenging to surmount; thus, pushing them lower on the priority list.

While this study focused on the compounds that were hits in both assays, this does not imply that the compounds that were hits in only one of the assays are not of interest. For example, NAC was a hit in the ototoxicity screen, but not the nephrotoxicity screen. There is significant overlap between the mechanisms of oto- and nephrotoxicity, including the role of elevated ROS levels, but also important differences, including specific drug transporter expression in the kidney vs. the cochlea (*Ciarimboli, 2012*; *Sprowl et al., 2013*). Thus, oto- and nephroprotective compounds do not necessarily share the same mechanism of protection. However, by studying two common toxicities caused by the same drug, we were able to identify compounds that could protect against both toxicities simultaneously, which would aid in avoiding polypharmacy in this already heavily treated population.

The multiple appearances of dopaminergic and serotonergic pathway modulators in this study (*Table 1*) is supported by similar findings in a study of both aminoglycoside- and cisplatin-induced zebrafish lateral line toxicity (*Vlasits et al., 2012*). Among the 10 hits emphasized in the Vlasits et al., study were paroxetine, a selective serotonin-reuptake inhibitor (SSRI) and fluspirilene, an antipsychotic compound that acts as an antagonist at the $D_2$ dopamine receptor (*Vlasits et al., 2012*). While these specific compounds did not show up as hits in the present screen, similar compounds were identified. For example, trifluoperazine dihydrochloride, a $D_2$-receptor antagonist was a hit in both assays in this study. Although this seems counterintuitive, the opposing effects of D1 vs. D2-

like receptors means that a D2-like receptor antagonist might have the same net effect as a D1-agonist. Of note, 15% of the Sigma LOPAC1280 library consists of compounds that would interact with either the dopaminergic or serotonergic pathway (9% and 6%, respectively). This is a relatively high proportion of compounds that could have translated to a disproportionate number of positive 'hits' in this pathway. It is also important to note that while L-mimosine may have dopaminergic effects, it also has a wide array of documented pharmacological effects, ranging from prolyl hydroxylase inhibition (*Mccaffrey et al., 1995*) to copper and iron chelation (*Kulp and Vulliet, 1996*).

The potential effects of dopamine and L-mimosine, in addition to other hits shown in *Figure 2c*, suggest that increased dopamine levels may be oto- and nephroprotective. However, further mechanistic experiments will need to be completed to support this hypothesis. D1 and D2-like receptors are both present in the mammalian inner ear (*Beaulieu and Gainetdinov, 2011*) and kidney (*Harris and Zhang, 2012*; *Jose et al., 1992*). Another point of interest is that organic anion transporter 2 (OCT2), the transporter that seems to play a role in cisplatin uptake into both renal tubule and cochlear cells, is also capable of transporting dopamine (*Ciarimboli et al., 2010*; *Filipski et al., 2008*; *Filipski et al., 2009*). Interestingly, there is also strong evidence that dopamine protects auditory nerves from glutamate-mediated excitotoxicity (*Lendvai et al., 2011*; *Ruel et al., 2001*). Although the hypothesis that oto- and nephroprotection may be mediated by increased dopamine signaling still needs to be confirmed, it is possible that dopamine could bind to the $D_1$ or $D_5$ receptors present in the inner ear and/or pronephros structure, causing increases in intracellular cAMP, which seems to provide nephroprotection in some studies (*Gillies et al., 2015*; *Hans et al., 1990*; *Palmieri et al., 1993*) and cochlear nerve protection from noise-induced hearing damage (*Darrow et al., 2007*; *Lendvai et al., 2011*; *Ruel et al., 2001*; *Oestreicher et al., 1997*).

L-DOPA (levodopa), the biosynthetic precursor to dopamine, is currently used to treat Parkinson's disease, which is mostly caused by the pathological loss of dopaminergic neurons in regions of the brain (*Salat and Tolosa, 2013*; *Sethi, 2010*). Individuals with Parkinson's disease have significantly worse hearing in comparison with age-matched controls (*Vitale et al., 2012*), a symptom of this neurodegenerative disorder that is partially improved by dopaminergic treatment (*Pisani et al., 2015*). In the present study, L-DOPA was a hit in the otoprotection assay, but not the nephroprotection assay, which may have been a result of only testing one dose at one time point in the initial screen; however, the potential utility of this clinically utilized compound should be the topic for further investigation.

In terms of potential nephroprotective effects, researchers have previously attempted to use low dose dopamine (0.5–3 µg/kg/min) to increase renal blood flow and potentially protect the kidneys from cisplatin (*Kellum, 1997*). The results of multiple studies were conflicting, and do not currently support the use of dopamine as a nephroprotectant (*Crona et al., 2017*). However, the inconsistency in findings may be a result of the presence of the antagonistic D1 and D2-like dopamine receptors in the kidney. One study found that fenoldopam, a selective $D_1$ receptor agonist reduced post-operative acute kidney injury (*Gillies et al., 2015*).

This study presents in vitro evidence in three different cell lines that the protective compounds do not interfere with the anticancer effects of cisplatin. While the reason for the specific protective effects of these compounds in human proximal tubule cells and zebrafish pronephros, neuromast and inner ear structures is unknown at this time, it is likely a result of intrinsic differences between the cells themselves, potentially at the level of dopamine receptor subtype expression. Thus, it will be essential to perform these in vivo experiments in a murine model prior to moving these compounds into human trials. These murine experiments will also be important, as despite being an excellent model organism for the study of oto- and nephrotoxicity, the zebrafish does not possess certain anatomical features that are present in mammals, like the stria vascularis or cochlea in the inner ear. Another important difference between human inner ear hair cells and zebrafish hair cells is that teleost hair cells (both lateral line and inner ear) regenerate, while mammalian hair cells do not (*Monroe et al., 2015*). Importantly in our studies, we did take care to ensure that regenerated hair cells were not captured in our experiments by rinsing each group of larvae separately and imaging or assessing them immediately.

Future experiments should focus on investigating the putative mechanisms of protection, which require further validation following the initial identification of these candidate protective compounds from our innovative screening strategy. Given the similarity in structure between dopamine and

L-mimosine, it would also be interesting to identify and assess similar chemical structures to potentially find more efficacious or potent compounds.

Despite advances in targeted chemotherapeutics and personalized medicine, broadly cytotoxic compounds like cisplatin remain a mainstay in the treatment of many malignancies. When cisplatin-induced toxicities are detected, clinicians are forced to decrease the dose or even discontinue this potentially life-saving drug. The present study developed a dual-purpose drug screen, identifying 22 compounds that showed potential as both oto- and nephroprotective. In particular, dopamine and L-mimosine were both oto- and nephroprotective in larval zebrafish. Further, in vitro studies in cancer cell lines demonstrated that dopamine and L-mimosine did not decrease the anticancer effects of cisplatin. These results demonstrate that dopamine and L-mimosine have significant promise as protective agents, warranting further investigation. These findings also suggest a potential biological role for elevated levels of dopamine in the protection against these toxicities. The drug screening methodology developed within this study could easily be applied to different libraries of compounds, either testing drugs for oto- or nephrotoxic effects, or for protection from these toxicities. Overall, the methods developed within this study, and the identification of several oto- and nephroprotective compounds, can hopefully inform future studies and improve the safety of cancer treatment.

# Materials and methods

## Key resources table

| Reagent type (species) or resource | Designation | Source or reference | Identifiers | Additional information |
|---|---|---|---|---|
| Cell line (*Homo sapiens*) | HK-2, proximal tubule cells, HPV-16 transformed | ATCC | CRL-2190, RRID:CVCL_0302 | |
| Cell line (*Homo sapiens*) | HSC-3, oral squamous cell carcinoma | JCRB Cell Bank | RRID:CVCL_1288 | |
| Cell line (*Homo sapiens*) | SK-N-AS, metastatic neuroblastoma cells | Gift from Dr. Meredith Irwin (SickKids) | CRL-2137, RRID:CVCL_1700 | |
| Cell line (*Homo sapiens*) | LAN5, metastatic neuroblastoma cells | Gift from Dr. Meredith Irwin (SickKids) | RRID:CVCL_0389 | |
| Fish strain (*Danio rerio*) | *Casper* double pigment mutant fish strain | Gift from Dr. Len Zon (Harvard), PMID:18371439 | $mitfa^{w2/w2};mpv17^{a9/a9}$ | Used throughout the fish work |
| Antibody | phospho-histone H2A.X (Ser139) (20E3) rabbit monoclonal antibody | Cell Signalling Technology | Cat # 9718S | IF: 1:400 |
| Chemical compound, drug | Cisplatin | Cayman Chemicals | Cat # 15663-27-1 | Prepared fresh in 0.9% NaCl before each experiment |
| Chemical compound, drug | Dopamine hydrochloride | Sigma-Aldrich | Cat # H60255 | Prepared fresh in 0.9% NaCl before each experiment |
| Chemical compound, drug | L-mimosine | Sigma-Aldrich | Cat # M0253 | Prepared in 0.9% NaCl, aliquoted and stored at −20°C |

## Zebrafish husbandry

Adult *casper* (*White et al., 2008*) zebrafish were housed in a recirculating commercial housing system (Pentair, Apopka, FL) at 28°C in 14 hr:10 hr light:dark conditions and bred according to standard protocol (*Westerfield, 1995*). *Casper* zebrafish were used throughout the study due to enhanced optical clarity and utility in fluorescence-based experimentation, without the need for phenylthiourea (PTU) treatment, which has been shown to influence both behavior and neural crest development (*Bohnsack et al., 2011*; *Parker et al., 2013*). Other groups have also used the *casper* mutant larvae to observe the zebrafish auditory system (*Wisniowiecki et al., 2016*) and demonstrate similar oto-toxicity responses in both *casper* mutants and wild-type larvae (d'*d'Alençon et al., 2010*). Embryos were collected and grown in E3 medium (5 mM NaCl, 0.17 mM KCl, 0.33 mM CaCl$_2$, 0.33 mM

$MgSO_4$) in an incubator maintained at 28°C. Embryos were cleaned daily, and provided with new E3 media, to remove debris and any embryos that died naturally. Zebrafish embryos (0–72 hpf) enter the free swimming larval stage after 72 hpf, and all larvae were used experimentally before seven dpf. The use of zebrafish in this study was approved by, and carried out in accordance with, the policies of the Dalhousie University Committee on Laboratory Animals (Protocols #17–131 and #17–055).

## Cell culture

Cells were maintained in standard cell culture conditions, with 5% $CO_2$ at 37°C. HK-2 proximal tubule cells were purchased from ATCC (American Type Culture Collection, Manassas, VA). HSC-3 oral squamous cell carcinoma cells were purchased from JCRB Cell Bank (Japanese Collection of Research Bioresources Cell Bank, Ibaraki, Osaka, Japan). SK-N-AS and LAN5 neuroblastoma (NBL) cell lines were a gift from Dr. Meredith Irwin (Sick Kid's, Toronto, ON, Canada). Cells were cultured according to their specific media requirements. Cell lines were authenticated and mycoplasma tested by IDEXX Bioanalytics (Columbia, MO).

## alamarBlue viability assays

Cell viability assays were performed using an alamarBlue (Thermo Fisher Scientific, Waltham, MA) assay, according to the manufacturer's instructions. Briefly, cells were incubated with alamarBlue at a final concentration of 10% for between 2.5 and 3 hr, then read on a plate reader with appropriate excitation and emission spectra (550/590 nm). Controls included culture media with drug alone, culture media with drug and alamarBlue, and cells with culture media with alamarBlue, but no drug. The negative control (culture media with drug and alamarBlue, but no cells) values were subtracted from experimental values. All values are expressed as % viability, in relation to the no drug control. For each replicate, all treatments were administered to at least two identical wells. All experiments were completed a minimum of three times.

## Drug treatments

To facilitate further use, the Sigma LOPAC1280 drug library (Sigma-Aldrich, St. Louis, MO) was transferred into 96-well plates, fitted with corresponding rubber lids. This library was used in both the in vitro and in vivo drug screens at a final concentration of 0.01 mM. Each drug screen was completed once, with either duplicate wells of HK-2 cells, or four larvae per well. These values were then averaged.

Cisplatin (Cayman Chemicals, Ann Arbor, MI) was prepared fresh for each experiment as a stock solution of 1.67 mM in 0.9% NaCl, protected from light. Dopamine hydrochloride (Sigma Aldrich) was prepared fresh for each experiment as a stock solution of 50 mM in 0.9% NaCl, protected from light. L-mimosine (Sigma Aldrich) was prepared as a 72 mM stock solution in 0.9% NaCl, aliquoted to prevent freeze-thaw cycles, and stored at −20°C.

In all experiments, zebrafish larvae were incubated with drugs at 35°C to more closely replicate human body temperature, representing a midpoint between the ideal temperature for zebrafish development (28°C) and human cells (37°C).

## Neuromast labeling

Larvae were pretreated with desired compound(s), depending on the specific experiment. Treatment was removed prior to the staining procedure. Neuromasts were stained by incubating larvae in 2 µM YO-PRO-1 Iodide 491–509 (Thermo Fisher Scientific) in methylene blue-free E3 media at 35°C for 1 hr. YO-PRO-1 is a carbocyanine nucleic acid stain that labels the nuclei of viable hair cells (*Baxendale and Whitfield, 2016*). Larvae were then rinsed 3X with methylene blue-free E3 media. Larvae were then analyzed using either an inverted Axio Observer Z1 microscope (Carl Zeiss, Oberkochen, Germany), or a Biosorter, coupled to a Large Particle Handler (Union Biometrica Inc, Holliston, MA).

## Biosorting

Larval biosorting was completed with a Biosorter (Union Biometrica Inc) with or without an associated Large Particle Handler (Union Biometrica Inc). When sampling from a 96-well plate, larvae were

arrayed with four larvae/well, and were exposed to 0.2 mg/ml tricaine immediately before sampling, to ensure linear passage of the larvae through the tubing. For these experiments, the unit was fitted with a 500 µM fluidics and optics core assembly (FOCA), 1000 µM fluid handling tubing, and a 488 nm laser. Although data were collected on all channels simultaneously, PH green fluorescence was examined in YO-PRO-1 studies, and overall green fluorescence was assessed in the GFR studies.

### Phalloidin inner ear hair cell staining

Staining of inner ear hair cells was performed as described by *Baxendale and Whitfield, 2016*. Briefly, larvae were treated at 60 hpf with either L-mimosine, dopamine, or a vehicle control. Larvae were rinsed at 72 hpf, then treated with 0.01 mM freshly-prepared cisplatin. At 120 hpf (48 hr cisplatin treatment), larvae were rinsed three times, euthanized with tricaine overdose, then fixed overnight at 4°C in 4% paraformaldehyde (PFA) in phosphate-buffered saline (PBS). Fixed larvae were washed 2X for 5 min in PBS+ 0.2% Triton-X100 (PBSTx), then permeabilized for 2 hr in PBS+2% Triton-X100. Following this, samples were washed 2X for 5 min in PBSTx, then incubated in Alexa Fluor 488 stock solution (diluted 1:20 in 1:1000 PBS-Tween), protected from light at room temperature (RT) for 1.5 hr. Samples were then washed 6X for 10 min in PBSTx, and stored at 4°C in the dark. Prior to imaging, samples were rinsed 4X for 15 min in PBS, then placed in a solution of 3:1 glycerol: Tris buffer (pH 8.1) containing 2% n-propyl gallate. Whole zebrafish larvae were mounted on standard slides with a 1.5-ounce cover slip, then viewed with a Zeiss LSM510 confocal microscope (Carl Zeiss), using Zeiss Zen2009 software. Specimens were epi-illuminated with a mercury lamp (X-Cite 120Q; Lumen Dynamics Inc) or with 30 mW argon (488 nm) laser (Carl Zeiss) directed through a 488/543 nm dichroic mirror (HFT 488/543; Carl Zeiss) onto the preparation. Emitted fluorescence was collected with a 25 × 0.80 NA objective (LCI Plan-Neofluar; Carl Zeiss) through a 505–530 nm filter (BP505-530; Carl Zeiss). Z-stacks were taken at regions of interest surrounding optic placode tissues and ranged from 27 to 42 µm in depth.

To ensure that differences in hair cell stereocilia appearance were not due to differences in fluorophore labeling, a standard scanning protocol was created within Zen and used for all specimens (2.0 digital zoom; 488 nm laser: 55% power, 500 master gain, 1.0 digital gain, 0.0 digital offset, 1.0 AU pinhole; set relative to control treatment groups). The experiment was repeated twice at minimum, with a minimum of three larval placodes on different larvae examined each time.

### Volumetric/3D assessment of hair cell ellipticity

The 3D structure of the hair cell stereocilia was reconstructed from laser scanning microscope (LSM) images using Imaris software v.X64 9.1.2 (Bitplane, Andor Technology, Oxford Instruments). The original Zeiss LSM files of 2D images ranging from 27 to 42 µm in depth were imported to Imaris software for surface reconstruction. This software was employed to obtain the topology of hair cells within the placode as well as overall hair cell morphology. Ellipticity (prolate, or elongation around the long axis) was used as a measure of the status of the hair cells with healthier hair cells exhibiting a higher prolate ellipticity. In order to analyze hair cell morphology, a total of 16 individual hair cells in six zebrafish from each of the treatment conditions were assessed. Any volumes outside of the placode, or hair cells which overlapped preventing individual quantification, were excluded from analysis.

### Inner ear hair cell enumeration

Fiji (ImageJ) was used to first create maximum intensity Z-projections of confocal images to count hair cells in the inner ear. These were conducted with the same images used for volumetric/ellipticity measurements. Individual hair cells were marked and counted using the Cell Counter plugin within ImageJ.

### In vivo glomerular filtration rate (GFR) assay

The in vivo assessment of GFR was adapted from experiments reported by *Hentschel et al., 2005*. Zebrafish larvae were treated at 60 hpf with either L-mimosine, dopamine, or a vehicle control. Larvae were rinsed at 72 hpf, then treated with 0.125 mM cisplatin. Larvae were rinsed three times with E3 media immediately before injection with FITC-inulin (Sigma Aldrich). A final concentration of 5% FITC-inulin was manually injected into the common cardinal vein using a PL1 Picoinjector (Harvard

Apparatus, Holliston, MA) of freshly rinsed treated zebrafish. Immediately following injection, larvae were screened under a fluorescent stereoscope for the presence of FITC in circulation to remove any uninjected larvae, then sorted using the Biosorter. Three representative larvae/group were then imaged using the Zeiss SteREO Discovery.V20 dissecting microscope (Carl Zeiss) then incubated in fresh E3 medium for 2 hr at 35°C, at which point the larvae were biosorted and imaged again. Data are presented as the fold change in overall green fluorescence as an indication of the glomerular filtration that occurred over the 2 hr rest period, with the 0 hr timepoint set to 1. The experiment was completed three times, with a minimum of 50 larvae measured/group.

## Histology

Larvae were treated in parallel with the GFR assay, then fixed overnight at 4°C in 4% paraformaldehyde (PFA) in PBS. Larvae were then pre-embedded in 1.5% Ultrapure low-melting point agarose (Sigma Aldrich), to allow for alignment of the larvae for easy coronal cross-sectioning. Pre-embedded larvae were fixed in 10% neutral buffered formalin for between 1 and 3 days, then embedded in Surgipath Paraplast (LEICA) with the Tissue Tek Embedding Station, Model TEC EMA-1 (Sakura Finetek USA Inc). Larvae were then sectioned to 4 μm with a Nikon Eclipse Ni manual microtome with Nikon Plan Fluor objectives (Nikon), and stained with H and E with a Tissue-Tek Prisma automated slide stainer. Slides were preserved with the Tissue-Tek Glas g2 automated cover-slipper (Sakura Finetek USA Inc). Slides were examined and imaged with the Nikon Digital Sight DS-L3 (Nikon).

## Flow cytometry

An Annexin V Apoptosis Detection kit (Thermo Fisher Scientific) was used for the detection of apoptotic cells. Cells were seeded in six-well plates and according to previously optimized doses and timelines, cells were pretreated with vehicle control or either 0.03 mM dopamine or L-mimosine for 12 hr. Cells were then treated with 0.01 mM freshly-prepared cisplatin. Cells ($1 \times 10^6$) were harvested with trypsin-EDTA (Thermo Fisher Scientific). Cells were washed 1X with PBS, 1X with Annexin V binding buffer, then were resuspended in binding buffer at a concentration of $1-5 \times 10^6$ cells/mL. PE-conjugated Annexin V (5 μl, excitation/emission = 488/561 nm) was added to 100 μl of the cell suspension and incubated for 15 min at RT in the dark. Samples were centrifuged and resuspended in 1 ml of Annexin binding buffer. SYTOXBlue dead cell stain (1 μL, excitation/emission = 444/480 nm) was added to each sample and incubated for 15 min at RT in the dark. Samples were placed on ice and subjected to flow cytometry using the BD FACSCanto flow cytometer. Data were analyzed using BD FACSDiva Software. Gating strategies for doublet discrimination and dead cell exclusion are demonstrated in *Figure 6—figure supplement 2a–i*.

## γ-H2AX labeling

Cells were plated similarly to the Annexin V apoptosis analysis, on sterile $18 \times 18$ mm glass coverslips (Globe Scientific, Mahwah, NJ) in six-well plates, then treated with the protective agents 12 hr prior to 0.01 mM cisplatin treatment. The following day, cells were rinsed once with PBS, then fixed with 4% PFA (in PBS) at 37°C for 20 min, then rinsed again with PBS. This time point was chosen as a result of a literature search that concluded that 24 hr following cisplatin administration is indicative of longer-term cytotoxicity (*Olive and Banáth, 2009*). Following fixation, cells were washed 3X with PBS, then permeabilized by incubating coverslips with 0.3% Triton-X-100 in PBS for 30 min at RT. Samples were then washed 3X with PBST. Samples were incubated with blocking buffer (5% donkey serum in PBST) for 1 hr at RT. In a dark humidity chamber, samples were incubated with phospho-histone H2A.X (Ser139) (20E3) rabbit monoclonal antibody (Cell Signalling Technology) at 1:400 in 2.5% donkey serum in PBST O/N at 4°C. The next day, samples were washed 4X with PBST, then were incubated in a dark humidity chamber with donkey polyclonal secondary antibody to rabbit IgG-H and L (Alexa Fluor 647) at 1:400 dilution, and 1:1000 DAPI (4',6-Diamidino-2-Phenylindole, Dihydrochloride) (Invitrogen) in 2.5% donkey serum in PBST for 1 hr at RT. Samples were then washed 4X with PBST. Following washing, coverslips were mounted with Dako Fluorescent mounting medium (Dako) onto Superfrost plus microslides (Thermo Fisher Scientific) and kept in the dark at RT overnight. Coverslips were stored at 4°C in the dark until imaging. Imaging of samples was done using the Zeiss LSM 710 Laser scanning confocal microscope. Four images were acquired for each treatment group/replicate. N = 3.

## Quantification of γH2AX staining

Quantification of γH2AX staining was completed with a custom ImageJ (Fiji) macro, written by Benno Orr (University of Toronto) (*Rueden et al., 2017*; *Schindelin et al., 2012*). For each confocal image, regions of interest (ROIs) for individual nuclei were generated from a thresholded duplicate of the DAPI channel. Thresholding was performed with the 'Triangle' method, noise was reduced using the default 'Despeckle' functionality, and holes within thresholded nuclei below a 500-pixel size cut-off and above a 0.5 circularity cut-off were filled. Non-overlapping nuclei were detected as thresholded particles above size and circularity cut-offs of 1000 pixels and 0.65, respectively. Nuclei with overlap were separated with the 'Watershed' functionality and detected as particles above a size and circularity cut-offs of 1000 pixels and 0.8, respectively. Area and integrated density in the DAPI and γH2AX channels were measured for each nucleus. Data are reported as γH2AX integrated density/DAPI integrated density, with each data point corresponding to an individual nucleus.

## Statistics and data processing

Data were accumulated in Microsoft Excel for Mac (Office 365 Version 16.22) and subsequently analyzed using Prism eight for MacOS (Version 8.1.2, GraphPad Software Inc). All data were tested for normal Gaussian distribution and parametric analyses were used when data were distributed normally and non-parametric tests were used when data were not normally distributed, when an appropriate replacement test was available. Unless otherwise mentioned, error bars represent standard deviation and experiments were completed three times. Statistics details can be found in *Supplementary file 2*.

Dose–response curves were assessed using methods defined by *Ritz et al., 2015*, using R software with the *drc* extension package. The graphs presenting the data show a scatterplot with the model represented with a blue line. The grey borders represent the 95% confidence intervals of the line. Briefly, this constitutes a four-parameter log-logistic model that takes into account the following parameters: *b,* the steepness of the dose-response curve; *c* and *d,* the lower and upper limits of the response; and *e,* the $ED_{50}$, or the effective dose at which half of maximal effect is observed (*Ritz et al., 2015*). The equation is as follows:

$$f(x, (b, c, d, e)) = c + \frac{d - c}{(1 + exp(b(log(x) - log(e))))}$$

The code is as follows:

$$Dose.response.model = drm(log.Response \sim Dose, data = Dose\_response, fct = LL.4())$$

$$summary(Dose.response.model)$$

# Acknowledgements

We acknowledge the Dalhousie Zebrafish Core Facility and The Cellular and Molecular Digital Imagine Core Facility for their contributions to this project. Special thanks go to Debra Wertman (University of British Columbia, BC, Canada) for her expertise and assistance in conducting the dose-response analysis. We also acknowledge Dr. Meredith Irwin (Sick Kids, ON, Canada) for kindly providing us with the neuroblastoma (NBL) cell lines.

# Additional information

### Competing interests

Shelby L Steele: Affiliated with Appili Therapeutics Inc. The author has no financial interests to declare. The other authors declare that no competing interests exist.

### Funding

| Funder | Grant reference number | Author |
| --- | --- | --- |
| Dalhousie University | Killam Predoctoral Award | Jaime N Wertman |

| IWK Health Centre | IWK Graduate Studentship | Jaime N Wertman |
|---|---|---|

No operating funds were directly associated with this work. Jaime Wertman was supported throughout the study by a Killam Predoctoral Award and an IWK Graduate Studentship. The funders had no role in study design, data collection and interpretation, or the decision to submit the work for publication.

## Author contributions

Jaime N Wertman, Conceptualization, Data curation, Formal analysis, Funding acquisition, Validation, Investigation, Visualization, Methodology, Writing - original draft, Writing - review and editing; Nicole Melong, Data curation, Supervision, Validation, Investigation, Methodology, Writing - review and editing; Matthew R Stoyek, Data curation, Formal analysis, Visualization, Writing - review and editing; Olivia Piccolo, Formal analysis, Investigation, Visualization, Methodology, Writing - review and editing; Stewart Langley, Validation, Investigation; Benno Orr, Formal analysis, Validation, Investigation, Visualization, Methodology, Writing - review and editing; Shelby L Steele, Supervision, Validation, Methodology; Babak Razaghi, Validation, Investigation, Methodology; Jason N Berman, Conceptualization, Resources, Data curation, Software, Supervision, Funding acquisition, Visualization, Writing - original draft, Project administration, Writing - review and editing

## Author ORCIDs

Jaime N Wertman (iD) https://orcid.org/0000-0001-6029-3376
Nicole Melong (iD) https://orcid.org/0000-0002-0429-0944
Jason N Berman (iD) https://orcid.org/0000-0002-4053-6067

## Ethics

Animal experimentation: The use of zebrafish in this study was approved by, and carried out in accordance with, the policies of the Dalhousie University Committee on Laboratory Animals (Protocols #17-131 and #17-055).

## Decision letter and Author response

Decision letter https://doi.org/10.7554/eLife.56235.sa1
Author response https://doi.org/10.7554/eLife.56235.sa2

# Additional files

## Supplementary files

• Supplementary file 1. Compounds from the Sigma LOPAC$^{1280}$ compound library that were toxic to ¾ zebrafish larvae at 0.01mM, in combination with 0.02mM cisplatin. Clinical usage information obtained from the ChEMBL database (*Gaulton et al., 2017*)

• Supplementary file 2. Statistical test details for manuscript.

• Transparent reporting form

## Data availability

Full list of hits from both the oto- and nephrotoxicity drug screens have been made available on Dryad (https://doi.org/10.5061/dryad.zcrjdfn8n).

The following dataset was generated:

| Author(s) | Year | Dataset title | Dataset URL | Database and Identifier |
|---|---|---|---|---|
| Wertman JN, Melong N, Stoyek MR, Piccolo O, Langley S, Orr B, Steele SL, Razaghi B, Berman JN | 2020 | Results from: The identification of dual protective agents against cisplatin-induced oto- and nephrotoxicity using the zebrafish model | https://doi.org/10.5061/dryad.zcrjdfn8n | Dryad Digital Repository, 10.5061/dryad.zcrjdfn8n |

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
