## [Decision Letter]

**Acceptance summary:**

This work describes, in a series of elegant experiments, an innovative approach for the robust identification of oto- and nephro-protective compounds that could be further investigated for the prevention of organ damage in cancer patients, mainly paediatric groups, receiving treatment with cisplatin.

**Decision letter after peer review:**

Thank you for submitting your article "The identification of dual protective agents against cisplatin-induced oto-and nephrotoxicity using the zebrafish model" for consideration by *eLife*. Your article has been reviewed by three peer reviewers, including Arduino A Mangoni as the Reviewing Editor and Reviewer #3, and the evaluation has been overseen by Richard White as the Senior Editor.

The reviewers have discussed the reviews with one another and the Reviewing Editor has drafted this decision to help you prepare a revised submission.

As the editors have judged that your manuscript is of interest, but as described below that additional experiments are required before it is published, we would like to draw your attention to changes in our revision policy that we have made in response to COVID-19 (https://elifesciences.org/articles/57162). First, because many researchers have temporarily lost access to the labs, we will give authors as much time as they need to submit revised manuscripts. We are also offering, if you choose, to post the manuscript to bioRxiv (if it is not already there) along with this decision letter and a formal designation that the manuscript is “in revision at *eLife*”. Please let us know if you would like to pursue this option. (If your work is more suitable for medRxiv, you will need to post the preprint yourself, as the mechanisms for us to do so are still in development.)

Summary:

This manuscript identifies compounds that prevent two common side-effects of cisplatin chemotherapy: ototoxicity and nephrotoxicity. The study uses an increasingly popular ototoxicity screening model, the larval zebrafish lateral line, and also employs a human kidney cell line for nephrotoxicity screening using the same drug library. This is a unique combination of two inter-related screens with the goal of identifying compounds that confer protection in both cell types. The study is clear and the findings are interesting; the screen hits include multiple modulators of dopaminergic synthesis and signalling. The authors also use a novel method (Biosorter) for lateral line screening, which increases the throughput of this assay. The nephrotoxicity studies in larval zebrafish are impressive – injecting into the cardinal vein of a larval fish is technically challenging. Also, the paper is written in a clear and logical manner.

Essential revisions:

– While the study is interesting, mechanistic data are lacking. The suggestion of dopaminergic modulation of oto- and nephrotoxicity is fascinating but there are no mechanistic data to demonstrate that their hits do indeed modulate dopamine signaling. Studies with dopamine receptor antagonists or knockout lines would present compelling mechanistic evidence. Furthermore, there is little mechanistic speculation in the Discussion.

– Some methodological choices are unclear. Why use 72 hpf animals, when the hair cells don't show mature-like responses until 96 hpf? How does cisplatin toxicity in the lateral line compare between casper mutants and wildtype animals? The casper mutants are likely needed for sell sorting, but melanophores can play a role in antioxidant defenses and therefore this mutant line may have altered responses to cisplatin. Finally, why were the zebrafish experiments performed at 35 C? Optimal temperature is 28 C and morbidity would be expected at this high temperature.

– In the zebrafish inner ear experiments, did the authors quantify cells in the saccule or utricle? The otic placode represents two distinct epithelia at this time point and these epithelia likely have different responses to ototoxic drugs (based on studies in adult fishes). Also, the ellipticity measurements new to the field and it's unclear if they accurately represent hair cell survival. Suggest comparing the ellipticity measurements to direct counts of phalloidin-labeled hair bundles, which are obvious in their images and relatively simple to quantify.

– There is considerable variability in hair cell survival in Figure 1A and in 2A for the cisplatin-only groups (red dots). This variability suggests that the automated Biosorter assay may not be as robust as the authors state. It would be useful to see a comparison of the Biosorter assay with a second method of lateral line hair cell quantification, such as counts of Yo-Pro-1 labeled cells (since they already use that dye), or hair cell counts using a hair cell marker such as anti-parvalbumin or a transgenic line such as Brn3c:mGFP or ET4. It would also be helpful to see a secondary validation method for the hair cell protection conferred by dopamine and L-mimosine (shown in Figure 3), since the images don't demonstrate robust increases in fluorescence.

– Even though the zebrafish possesses hair cells and a renal system, the authors should also underline what are the limitations of the comparison with mammal hair cells and kidneys. A brief description of these structures would be helpful for readers not confident with zebrafish anatomy and physiology.

– It was stated that: “…Treatment with any of these compounds may result in increased synaptic dopamine levels….” Since these compounds protected also HK2 cells, which do not contain synapsis, this suggestion is not appropriate.

– Measurement of GFR in zebrafish: did the authors measure the effect of dopamine or mimosine alone on GFR?

– The authors speculate that the protection against cisplatin toxicities may be somehow associated with the characteristics of the protective substances as member of dopaminergic and serotonergic transmission. However, trifluoperazine dihydrochloride is a D2-receptor antagonist and dopamine, of course, an agonist of D-receptors. For this reason, the association between dopaminergic actions and protection is unclear.

– One of the possible explanations of cisplatin oto- and nephrotoxicity is a specific interaction of cisplatin with organic cation transporters (OCTs), which are highly expressed in the sites of cisplatin unwanted toxicities. Since dopamine is a well-known substrate of OCTs, it is unclear why the authors did not comment this. There are studies demonstrating the expression of OCTs in the zebrafish kidneys and hair cells. This may be an important point to be discussed.

– Please clarify whether the cisplatin dose range in neuromast and kidney tubule cells toxicity studies resembles the plasma/serum concentrations measured in paediatric cancer patients.

[Editors' note: further revisions were suggested prior to acceptance, as described below.]

Thank you for re-submitting your article "The identification of dual protective agents against cisplatin-induced oto-and nephrotoxicity using the zebrafish model" for consideration by *eLife*. Your article has been re-reviewed by two peer reviewers, and the evaluation has been overseen by a Reviewing Editor and Richard White as the Senior Editor.

While the manuscript has been improved, there are still some issues remaining:

Essential revisions:

– The manuscript has been strongly improved. Even though an elucidation of the mechanisms explaining the observed protection is beyond the aims of this work, there is no mechanistic comment on the missing protection by dopamine and mimosine of cancer cells: the treatment of these cells with these substances did not change or even increased cisplatin cellular toxicity. It would be of interest to discuss a little bit this point.

– I appreciate the authors adding some additional information about dopamine receptors. I would like an additional sentence or two tying their findings with dopamine and L-mimosine to what is known about dopaminergic signaling in the auditory periphery; why might increasing dopamine be protective?

– I still have concerns about the 35 C incubation temperature; zebrafish metabolism evolved to operate effectively at lower temperatures, not at human body temperature. I recognize that all experiments were conducted at 35 C and do not suggest rerunning experiments for the present manuscript, but I caution the authors to use species-specific temperatures for their future work.

---

## [Author Response]

Essential revisions:– While the study is interesting, mechanistic data are lacking. The suggestion of dopaminergic modulation of oto- and nephrotoxicity is fascinating but there are no mechanistic data to demonstrate that their hits do indeed modulate dopamine signaling. Studies with dopamine receptor antagonists or knockout lines would present compelling mechanistic evidence. Furthermore, there is little mechanistic speculation in the Discussion.

The authors agree with the reviewers that mechanistic studies are lacking. However, these mechanistic studies were felt to be beyond the scope of this manuscript at the current time. We agree that some of the language in the manuscript needs to be tempered to reflect the hypothetical nature of this suggestion.

These changes are highlighted throughout the manuscript and include the following:

“The potential effects of dopamine and L-mimosine, in addition to other hits shown in Figure 2C, suggest that increased dopamine levels may be oto- and nephroprotective. However, further mechanistic experiments will need to be completed to support this hypothesis.”

“Future experiments should focus on investigating the putative mechanisms of protection, which require further validation following the initial identification of these candidate protective compounds from our innovative screening strategy.”

“These findings also suggest a potential biological role for elevated levels of dopamine in the protection against these toxicities.”

– Some methodological choices are unclear. Why use 72 hpf animals, when the hair cells don't show mature-like responses until 96 hpf?

We appreciate the thoughtfulness of this question. While we considered beginning these experiments at later time points, we ultimately decided to begin experiments at 3 days post-fertilization (dpf) because we were conducting a relatively long-term treatment with cisplatin (48hrs) and it is part of our standard operating protocol to begin feeding larvae at 5 dpf. Please note that this is to ensure that there are fewer uncontrolled variables and that the food does not interfere with imaging or fluorescence. Furthermore, other similar studies begin short treatment at 4 or 5 days (Chiu et al., 2008; Harris et al., 2003), acknowledging that lateral line hair cells are present at 3 dpf (Harris et al., 2003). Thus, with our longer-term treatment, the end point of these experiments is similar to ours.

How does cisplatin toxicity in the lateral line compare between casper mutants and wildtype animals? The casper mutants are likely needed for sell sorting, but melanophores can play a role in antioxidant defenses and therefore this mutant line may have altered responses to cisplatin.

While we had originally planned to use wild type larvae, the reduced pigmentation afforded by *casper* double pigment mutants was needed for biosorting to measure both lateral line health (with YO-PRO1 staining) and pronephros filtration (with the inulin-based glomerular filtration assay). It also allowed for enhanced imaging opportunities for the phalloidin-based inner ear hair cells, without the need for treatment with a clearing agent. We have added a line in the Materials and methods section to reflect this choice. It reads as follows:

“*Casper* zebrafish were used throughout the study due to enhanced optical clarity and utility in fluorescent-based experimentation.”

While we have not specifically quantified the differences between *casper* vs. wild type larvae responses to ototoxins, studies have found similar leukocyte recruitment/inflammatory responses within the lateral line system following ototoxin insult in both *casper* and wild type larvae (d’Alençon et al., 2010). Other groups have also taken advantage of the optical transparency of *casper* larvae to further explore the auditory system in zebrafish (Wisniowiecki et al., 2016). Some studies have also warned against the use of PTU, as it can influence both behaviour (Parker et al., 2013) and neural crest formation (Bohnsack, Gallina and Kahana, 2011). As a result, we made the decision to work with the *casper* mutant larvae.

Finally, why were the zebrafish experiments performed at 35 C? Optimal temperature is 28 C and morbidity would be expected at this high temperature.

The zebrafish experiments were intentionally performed at 35°C, as this is closer to human body temperature. We have found through our toxicity assays that zebrafish respond differently to drugs at various temperatures. Since zebrafish larvae can tolerate being maintained at 35°C, as we have established with our xenograft studies (Bentley et al., 2014; El-Naggar et al., 2015; Liu et al., 2014; Melong et al., 2017; Pringle et al., 2019; Veinotte, Dellaire and Berman, 2014)), we decided to perform drug-based experiments at this temperature, that more closely replicates human body temperature. It should also be noted that the untreated control zebrafish larvae were also maintained at 35°C for consistency. We note that we did not specify this in the Materials and methods and this statement has since been added:

“In all experiments, zebrafish larvae were incubated with drugs at 35°C to more closely replicate the temperature of the human body, representing a midpoint between the ideal temperature for zebrafish development (28°C) and human cells (37°C).”

– In the zebrafish inner ear experiments, did the authors quantify cells in the saccule or utricle? The otic placode represents two distinct epithelia at this time point and these epithelia likely have different responses to ototoxic drugs (based on studies in adult fishes). Also, the ellipticity measurements new to the field and it's unclear if they accurately represent hair cell survival. Suggest comparing the ellipticity measurements to direct counts of phalloidin-labeled hair bundles, which are obvious in their images and relatively simple to quantify.

We agree with the reviewers that basic counts of hair cell survival would help support the ellipticity measurements in Figure 4. We have performed this analysis on the same images that were used for the ellipticity measurements and have reported them in the manuscript as a new panel of this figure, replacing one of the ellipticity measurements (Figure 4F). We would like to point out that the overall trend between hair cell number is the same as ellipticity measurement. We have amended the Results section to reflect this, as follows:

“In order to ensure that the ellipticity measurements reflected the same trends that we would see with overall hair cell number, we used the Cell Counter plugin for ImageJ to quantify the number of hair cells in the same maximum-projection z-stack images that were used for Imaris analysis (Figure 4F). It should be noted that the exact same trend is observed, where dopamine and L-mimosine pretreatments resulted in hair cell numbers that were higher than those observed in larvae treated with cisplatin alone.”

We have also amended the Materials and methods section to include this additional analysis.

We imaged the larvae with the following orientation – anterior to the left and dorsal at the top, as a lateral orientation of the zebrafish to the objective gave the most unobstructed view into the inner ear. This means that we were imaging the posterior macula. At this time point (6 dpf), there are two main macula – the anterior and posterior. The posterior macula is thought to give rise to the saccule. This orientation allowed for optical sectioning through the hair cells transversely, allowing more slices per hair cell to get a high-quality 3D reconstruction. If the same lateral orientation was used, it would be necessary to optically section any other placode longitudinally, reducing the number of slices per cell, ultimately reducing resolution of imaging. Attempting to adjust the orientation of the larvae to get transverse sections of the other placodes resulted in interference by bone and muscle tissues.

It should also be noted that, while differential organ sensitivities may be observed in adult zebrafish, this has not been closely studied in larvae. However, the posterior macula is thought to give rise to the saccule, which is a region that is investigated in adult zebrafish studies of related phenomenon, like noise-induced hearing loss (Monroe, Rajadinakaran and Smith, 2015).

– There is considerable variability in hair cell survival in Figure 1A and in 2A for the cisplatin-only groups (red dots). This variability suggests that the automated Biosorter assay may not be as robust as the authors state. It would be useful to see a comparison of the Biosorter assay with a second method of lateral line hair cell quantification, such as counts of Yo-Pro-1 labeled cells (since they already use that dye), or hair cell counts using a hair cell marker such as anti-parvalbumin or a transgenic line such as Brn3c:mGFP or ET4. It would also be helpful to see a secondary validation method for the hair cell protection conferred by dopamine and L-mimosine (shown in Figure 3), since the images don't demonstrate robust increases in fluorescence.

We agree with the reviewers that there is variability in hair cell staining quantification displayed in Figure 1A and 2A. This degree of variability was fairly consistent across experiments. While this could be viewed as a potential weakness, we believe that this shows one of the strengths of the zebrafish model, namely the opportunity for the large number of animals available in zebrafish larvae experiments to capture some of the natural variability that may exist within a given population, with a large enough sample size to still detect meaningful differences.

That said, during the validation of the biosorter for this method, we did attempt several other approaches to determine the impact of cisplatin treatment on lateral line integrity. While other studies have employed manual counting of hair cell bundles, we saw this as cumbersome and prone to human error, especially seeing as neuromast hair cell bundles often contain living and dead hair cells simultaneously – making strict +/- counting challenging. We did assess neuromast integrity following YO-PRO1 staining in two different methods. First, we used built-in software on our fluorescent microscope (ZEN2, Blue Edition) to measure the change in neuromast fluorescence. While this method was able to detect a decrease in neuromast fluorescence following cisplatin treatment, it involved imaging each larva independently and was very time consuming. Before acquiring the biosorter, we also attempted to make the neuromast assay more high throughput by measuring the fluorescence of treated YO-PRO1 stained larvae in a 96-well plate format with a fluorescent plate reader. While this method was able to detect differences between treated vs. untreated larvae, it did not detect differences between larvae that were treated with two different doses of cisplatin, possibly limiting its utility. Please see the results in Author response image 1 and 2.

**Author response image 1. sa2fig1:** EFCisplatin treatment results in damage to zebrafish neuromast structures that is detectable with ZEN software. Groups of 5 x 72 hour post-fertilization (hpf) casperzebrafish larvae were treated with either vehicle control, or 0.02mM freshly-prepared cisplatin for 24hr. At the endpoint, larvae were stained with 2μM YO-PRO1 for 1hr, rinsed, then imaged with fluorescent microscopy. ZEN software was used to analyze the average level of green fluorescence in each image. Data is represented as fold change in comparison to the untreated control. A) Analysis of fluorescence values at 5X magnification. Each data point represents an individual larvae. **=p=0.011, as per two-tailed student’s T-test. B-C) Representative larvae and fluorescent histogram for B) untreated control, and C) cisplatin treated larvae. D) Analysis of fluorescence values at 10X magnification. Each data point represents an individual larvae. ****=p<0.0001, as per two-tailed student’s T-test. E-F) Representative larvae and fluorescent histogram for E) untreated control, and F) cisplatin treated larvae. N=2 replicates with 5 larvae each. Error bars represent standard deviation (SD).

**Author response image 2. sa2fig2:** Cisplatin treatment results in damage to zebrafish neuromast structures that is detectable using a plate reader. Groups of 72 hour post-fertilization (hpf) casperzebrafish larvae were treated with either vehicle control, 0.05 or 0.25mM cisplatin for 24hr. At the endpoint, larvae were stained with 2μM YO-PRO1 for 1hr, rinsed, then allotted 5X well in a 96-well plate. Fluorescence values were then acquired using a plate reader. Data is presented as fold change in comparison with untreated controls. Each data point represents the average of 3 reads of an individual well that contained 5 larvae. ****=p<0.0001, as per one-way ANOVA with a Tukey post-test. N=2 replicates with at least 4 wells in each treatment.

As suggested by the reviewers, we originally attempted to utilize a transgenic reporter fish line, like the Brn3c:mGFP; however, we did not have access to this line due to difficulties in the importation of transgenic fish into Canada at the time of experimentation. Furthermore, it has been contended that some reporter lines, like Brn3c:mGFP, have differing hearing sensitivity in comparison with their wild-type counterparts (Monroe et al., 2016). In order to stay consistent, we decided to conduct this work within the *casper* line, due to its inherent facility for visualizing fluorescent labeling.

– Even though the zebra fish possesses hair cells and a renal system, the authors should also underline what are the limitations of the comparison with mammal hair cells and kidneys. A brief description of these structures would be helpful for readers not confident with zebrafish anatomy and physiology.

We agree with the reviewers that a description of the limitations of using the zebrafish as a model for mammalian inner ears and kidneys would enhance the manuscript. We have added some more detail about their structure in the Introduction and a section in the Discussion that addresses the shortcomings of this model. In particular, we mention the fact the absence of certain physical structures that may modulate hair cell function, like the stria vascularis and cochlea.

The wording that was added to the Introduction. It now reads as follows: “Zebrafish larvae as young as four days post-fertilization (dpf) possess a functional pronephros structure (a primitive kidney that resembles the two nephrons, running along the length of the larva) and a sophisticated oto-vestibulary system including an inner ear-like structure with multiple sensory placodes (Baxendale and Whitfield, 2016; Drummond and Davidson, 2010; Hentschel et al., 2005; Whitfield et al., 2002).”

The section that was inserted into the Discussion is as follows:

“These murine experiments will also be important, as despite being an excellent model organism for the study of oto- and nephrotoxicity, the zebrafish does not possess certain anatomical features that are present in mammals, like the stria vascularis or cochlea in the inner ear.”

Another important difference between zebrafish hair cells (both inner ear and lateral line) and human inner ear hair cells is that teleost hair cells regenerate, and mammalian hair cells do not. This will be important for future work, which will hopefully involve an investigation into the mechanism of protection, but was also an important consideration for experimental design. Zebrafish hair cells have been observed to begin regenerating (from supporting cells) as early as 24 hr post damage, although this length of time seems to extend with increased levels of damage (Hernandez et al., 2007). To ensure that regeneration did not interfere with our experiments, we ensured that all groups of larvae were treated for the same length of time with cisplatin and were imaged/assessed/sacrificed and fixed the same length of time after rinsing. This meant that we rinsed each group individually, then measured it immediately, to avoid capturing any potential regeneration. A sentence has been added to the Discussion to reflect this important detail.

“Another important difference between human inner ear hair cells and zebrafish hair cells is that teleost hair cells (both lateral line and inner ear) regenerate, while mammalian hair cells do not (Monroe et al., 2015). Importantly in our studies, we did take care to ensure that regenerated hair cells were not captured in our experiments by rinsing each group of larvae separately and imaging or assessing them immediately. “

– It was stated that: “…Treatment with any of these compounds may result in increased synaptic dopamine levels….” Since these compounds protected also HK2 cells, which do not contain synapsis, this suggestion is not appropriate.

We appreciate the reviewers pointing out this oversight. This section of the text has been amended to read:

“Treatment with any of these compounds may result in increased dopamine signalling.”

– Measurement of GFR in zebrafish: did the authors measure the effect of dopamine or mimosine alone on GFR?

No, we did not measure the effects of dopamine or L-mimosine alone on zebrafish GFR. Although we wanted to complete this experiment, we were unable to do so for technical reasons. As the reviewers noted in their summary, common cardinal vein injection is technically challenging and time consuming. Operation of the biosorter is also a technical task that requires special training. Since we used the biosorter to measure larval fluorescence immediately after injection and at 2 hours post-injection, the timeline made adding additional groups into the experiment impracticable.

Although we were not able to test the effects of the protective agents in vivo in the absence of cisplatin, we did measure the impact of the protective agents on HK-2 human proximal tubule cell viability. At both 24 hours and 48 hours post-treatment, there was no change in viability of HK-2 cells with either 0.01 or 0.03mM protective agent pretreatment.

– The authors speculate that the protection against cisplatin toxicities may be somehow associated with the characteristics of the protective substances as member of dopaminergic and serotonergic transmission. However, trifluoperazine dihydrochloride is a D2-receptor antagonist and dopamine, of course, an agonist of D-receptors. For this reason, the association between dopaminergic actions and protection is unclear.

We agree that this type of claim on the surface may sound counterintuitive, but the action of a D2-like receptor antagonist may have similar activity as dopamine itself. Generally speaking, the D1-like class of receptors activate Gα_s/olf_ family of G proteins, stimulating production of cyclic AMP (cAMP) by adenylyl cyclase (AC). The D2-like class couple with the Gα_i/o_ family of G proteins, resulting in the inhibition of AC, decreasing cAMP production. Thus, a D2-like antagonist could have the same net effect of increasing cAMP production as a D1-like receptor agonist. However, it should also be noted that dopamine itself is a “dirty” neurotransmitter – meaning it can activate both D1 and D2-like dopamine receptor classes (Beaulieu and Gainetdinov, 2011). We also recognize that this study did not undertake any mechanistic analysis, so as mentioned above, we have attempted to temper the language surrounding these claims.

We have also inserted a sentence in the Discussion that addresses this:

“Although this seems counterintuitive, the opposing effects of D1 vs. D2-like receptors means that a D2-like receptor antagonist might have the same net effect as a D1-agonist.”

– One of the possible explanations of cisplatin oto- and nephrotoxicity is a specific interaction of cisplatin with organic cation transporters (OCTs), which are highly expressed in the sites of cisplatin unwanted toxicities. Since dopamine is a well-known substrate of OCTs, it is unclear why the authors did not comment this. There are studies demonstrating the expression of OCTs in the zebrafish kidneys and hair cells. This may be an important point to be discussed.

We agree with the reviewers. This fact has been added to the Discussion, as it definitely represents an area of interest.

The section that has been added is as follows:

“Another point of interest is that organic anion transporter 2 (OCT2), the transporter that seems to play a role in cisplatin uptake into both renal tubule and cochlear cells, is also capable of transporting dopamine (Ciarimboli et al., 2010; Filipski et al, 2008; Filipski et al., 2009).”

– Please clarify whether the cisplatin dose range in neuromast and kidney tubule cells toxicity studies resembles the plasma/serum concentrations measured in paediatric cancer patients.

This is an excellent question and we would like to thank the reviewers for raising this issue. Studies of serum cisplatin concentrations suggest that levels peak quickly after administration, then decline rapidly (Rajkumar et al., 2016). However, human studies suggest that over the initial 24 hour period following administration, plasma levels range from approx. 1-10 µg/mL, which translates to 0.0033 mM-0.033mM. This is within the range of what we used experimentally.

We have added a line in the Discussion to emphasize this:

“Human studies suggest that over the initial 24 hr period following administration, plasma cisplatin levels range from approx. 1-10 µg/mL, which translates to 0.0033 mM-0.033mM (Lanvers-Kaminsky et al., 2006; Rajkumar et al., 2016), well within the range of what we used experimentally.”

[Editors' note: further revisions were suggested prior to acceptance, as described below.]

Essential revisions:– The manuscript has been strongly improved. Even though an elucidation of the mechanisms explaining the observed protection is beyond the aims of this work, there is no mechanistic comment on the missing protection by dopamine and mimosine of cancer cells: the treatment of these cells with these substances did not change or even increased cisplatin cellular toxicity. It would be of interest to discuss a little bit this point.

We agree with the reviewers that this should be mentioned in the manuscript. Although we do not know the reason for the differences in protection (between non-cancer vs. cancer cells) at this time, we have added in a suggestion that the differences likely are a result of intrinsic cell-based differences. This now reads as follows:

“This study presents in vitro evidence in three different cell lines that the protective compounds do not interfere with the anticancer effects of cisplatin. While the reason for the specific protective effects of these compounds in human proximal tubule cells and zebrafish pronephros, neuromast and inner ear structures is unknown at this time, it is likely a result of intrinsic differences between the cells themselves, potentially at the level of dopamine receptor subtype expression.”

– I appreciate the authors adding some additional information about dopamine receptors. I would like an additional sentence or two tying their findings with dopamine and L-mimosine to what is known about dopaminergic signaling in the auditory periphery; why might increasing dopamine be protective?

We have added a sentence in the Discussion addressing this notion.

“Although the hypothesis that oto- and nephroprotection may be mediated by increased dopamine signaling still needs to be confirmed, it is possible that dopamine could bind to the D_1_ or D_5_ receptors present in the inner ear and/or pronephros structure, causing increases in intracellular cAMP, which seems to provide nephroprotection in some studies (Gillies et al., 2015; Hans et al., 1990; Palmieri et al., 1993) and cochlear nerve protection from noise-induced hearing damage (Darrow et al., 2007; Lendvai et al., 2011; Nouvian et al., 2001; Oestreicher et al., 1997).”

– I still have concerns about the 35 C incubation temperature; zebrafish metabolism evolved to operate effectively at lower temperatures, not at human body temperature. I recognize that all experiments were conducted at 35 C and do not suggest rerunning experiments for the present manuscript, but I caution the authors to use species-specific temperatures for their future work.

We thank the reviewers for this comment, and we both appreciate and understand this concern. This will be discussed further before proceeding with future studies.

**References**

Baxendale, S., and Whitfield, T. T. (2016). Methods to study the development, anatomy, and function of the zebrafish inner ear across the life course. Methods in Cell Biology, 134, 165–209.

Beaulieu, J., and Gainetdinov, R. R. (2011). The Physiology, Signaling, and Pharmacology of Dopamine Receptors. Pharmacological Reviews, 63(1), 182–217. https://doi.org/10.1124/pr.110.002642.182

Bentley, V. L., Veinotte, C. J., Corkery, D. P., Pinder, J. B., LeBlanc, M. A., and Bedard, K. (2014). Focused chemical genomics using zebrafish xenotransplantation as a preclinical therapeutic platform for T-cell acute lymphoblastic leukemia. Haematologica.

Bohnsack, B. L., Gallina, D., and Kahana, A. (2011). Phenothiourea sensitizes zebrafish cranial neural crest and extraocular muscle development to changes in retinoic acid and IGF signaling. PLoS ONE, 6(8). https://doi.org/10.1371/journal.pone.0022991

Brock, Knight, K. R., Freyer, D. R., Campbell, K. C. M., Steyger, P. S., Blakley, B. W., … Neuwelt, E. A. (2012). Platinum-induced ototoxicity in children: A consensus review on mechanisms, predisposition, and protection, including a new International Society of Pediatric Oncology Boston ototoxicity scale. Journal of Clinical Oncology, 30(19), 2408–2417. https://doi.org/10.1200/JCO.2011.39.1110

Brock, P. R., Koliouskas, D. E., Barratt, T. M., Yeomans, E., and Pritchard, J. (1991). Partial reversibility of cisplatin nephrotoxicity in children. The Journal of Pediatrics, 118(4, Part 1), 531–534. https://doi.org/https://doi.org/10.1016/S0022-3476(05)83372-4

Chiu, L., Cunningham, L., Raible, D., Rubel, E., and Ou, H. (2008). Using the Zebrafish Lateral Line to Screen for Ototoxicity. JARO - Journal of the Association for Research in Otolaryngology, 9, 178–190. https://doi.org/10.1007/s10162-008-0118-y

Ciarimboli, G., Deuster, D., Knief, A., Sperling, M., Holtkamp, M., Edemir, B., … Koepsell, H. (2010). Organic Cation Transporter 2 Mediates Cisplatin-Induced Oto- and Nephrotoxicity and Is a Target for Protective Interventions. The American Journal of Pathology, 176(3), 1169–1180. https://doi.org/10.2353/ajpath.2010.090610

Crona, D., Faso, A., Nishijima, T., McGraw, K., Galsky, M., and Milowshy, M. (2017). A Systematic Review of Strategies to Prevent Cisplatin-Induced Nephrotoxicity. The Oncologist, 22, 609–619.

d’Alençon, C. A., Peña, O. A., Wittmann, C., Gallardo, V. E., Jones, R. A., Loosli, F., … Allende, M. L. (2010). A high-throughput chemically induced inflammation assay in zebrafish. BMC Biology, 8. https://doi.org/10.1186/1741-7007-8-151

Ding, D., Liu, H., Qi, W., Jiang, H., Li, Y., Wu, X., … Salvi, R. (2016). Ototoxic effects and mechanisms of loop diuretics. Journal of Otology, 11(4), 145–156. https://doi.org/10.1016/j.joto.2016.10.001

Drummond, I. A., and Davidson, A. J. (2010). Zebrafish kidney development. Methods in Cell Biology (Third Edit, Vol. 100). Elsevier Inc https://doi.org/10.1016/B978-0-12-384892-5.00009-8

El-Naggar, A., Veinotte, C., Tognon, C., Corkery, D., Cheng, H., Tirode, F., … Sorensen, P. (2015). Translational activation of HIF1a by YB-1 promotes sarcoma metastasis. Cancer Cell, 27(5), 682–697.

Filipski, K. K., Loos, W. J., Verweij, J., and Sparreboom, A. (2008). Interaction of Cisplatin with the Human Organic Cation Transporter 2. Clinical Cancer Research, 14(12), 3875–3881. https://doi.org/10.1158/1078-0432.CCR-07-4793

Filipski, K. K., Mathijssen, R. H., Mikkelsen, T. S., Schinkel, A. H., and Sparreboom, A. (2009). Contribution of organic cation transporter 2 (OCT2) to cisplatin-induced nephrotoxicity. Clin Pharmacol Ther, 86(4), 396–402. https://doi.org/10.1038/clpt.2009.139.Contribution

Harris, J. A., Cheng, A. G., Cunningham, L. L., MacDonald, G., Raible, D. W., and Rubel, E. W. (2003). Neomycin-Induced Hair Cell Death and Rapid Regeneration in the Lateral Line of Zebrafish (*Danio rerio*). Journal of the Association for Research in Otolaryngology, 234, 219–234. https://doi.org/10.1007/s10162-002-3022-x

Hayes, D., Cvitkovic, E., Golbey, R., Scheiner, E., Helson, L., and Krakoff, I. (1977). High dose cis-platinum diammine dichloride: amelioration of renal toxicity by mannitol diuresis. Cancer, 39, 1372–1381.

Hentschel, D. M., Park, K. M., Cilenti, L., Zervos, A. S., Drummond, I., Bonventre, J. V, … Antonis, S. (2005). Acute renal failure in zebrafish: a novel system to study a complex disease. Am J Physiol Renal Physiol, 288, 923–929. https://doi.org/10.1152/ajprenal.00386.2004.

Hernandez, Pedro, P., Olivari, F. A., Sarrazin, A. F., Sandoval, P. C., and Allende, M. L. (2007). Regeneration in zebrafish lateral line neuromasts: expression fo the neural progenitor cell marker Sox2 and proliferation-dependent and independent mechanisms of hair cell renewal. Developmental Neurobiology, 67, 637–654. https://doi.org/10.1002/dneu

Lanvers-Kaminsky, C., Krefeld, B., Dinnesen, A., Deuster, D., Seifert, E., Wurthwein, G., … Boos, J. (2006). Continuous or repeated prolonged cisplatin infusions in children: a prospective study on ototoxicity, platinum concentrations, and standard serum parameters. Pediatric Blood and Cancer, 47, 183–193. https://doi.org/10.1002/pbc

Liu, Y., Asnani, A., Zou, L., Bentley, V. L., Yu, M., Wang, Y., … Peterson, R. T. (2014). Visnagin protects against doxorubicin-induced cardiomyopathy through inhibtion of mitochondrial malate dehydrogenase. Science Translational Medicine, 6(266), 266ra170-266ra170. https://doi.org/10.1126/scitranslmed.3010189

Melong, N., Steele, S., MacDonald, M., Holly, A., Collins, C. C., Zoubeidi, A., … Dellaire, G. (2017). Enzalutamide inhibits testosterone-induced growth of human prostate cancer xenografts in zebrafish and can induce bradycardia. Scientific Reports, 7(1), 14698.

Monroe, J. David, Manning, D. P., Uribe, P. M., Bhandiwad, A., Sisneros, J. A., Smith, M. E., and Coffin, A. B. (2016). Hearing sensitivity differs between zebrafish lines used in auditory research. Hearing Research, 341, 220–231. https://doi.org/10.5993/AJHB.40.1.1.The

Monroe, Jerry D., Rajadinakaran, G., and Smith, M. E. (2015). Sensory hair cell death and regeneration in fishes. Frontiers in Cellular Neuroscience, 9(APR), 1–18. https://doi.org/10.3389/fncel.2015.00131

Parker, M. O., Brock, A. J., Millington, M. E., and Brennan, C. H. (2013). Behavioural phenotyping of casper mutant and 1-Pheny-2-Thiourea treated adult zebrafish. Zebrafish, 10(4), 466–471. https://doi.org/10.1089/zeb.2013.0878

Pringle, E., Wertman, J., Melong, N., Coombs, A., Young, A., O’Leary, D., … Berman, J. (2019). The zebrafish xenograft platform - A novel tool for modeling KSHV-associated diseases. Preprints, 2019110081. https://doi.org/1020944/preprints201911.0081.v1

Rajkumar, P., Mathew, B. S., Das, S., Isaiah, R., John, S., Prabha, R., and Fleming, D. H. (2016). Cisplatin concentrations in long and short duration infusion: Implications for the optimal time of radiation delivery. Journal of Clinical and Diagnostic Research, 10(7), XC01–XC04. https://doi.org/10.7860/JCDR/2016/18181.8126

Santoso, J. T., Lucci, J. A., Coleman, R. L., Schafer, I., and Hannigan, E. V. (2003). Saline, mannitol, and furosemide hydration in acute cisplatin nephrotoxicity: a randomized trial. Cancer Chemotherapy and Pharmacology, 52, 13–18. https://doi.org/10.1007/s00280-003-0620-1

Veinotte, C. J., Dellaire, G., and Berman, J. N. (2014). Hooking the big one: the potential of zebrafish xenotransplantation to reform cancer drug screening in the genomic era. Disease Models and Mechanisms, 7(7), 745–754. https://doi.org/10.1242/dmm.015784

Viglietta, V., Shi, F., Hu, Q. Y., Ren, Y., Keilty, J., Wolff, H., … Soglia, J. (2020). Phase 1 study to evaluate safety, tolerability and pharmacokinetics of a novel intra-tympanic administered thiosulfate to prevent cisplatin-induced hearing loss in cancer patients. Investigational New Drugs. https://doi.org/10.1007/s10637-020-00918-1

Whitfield, T. T., Riley, B. B., Chiang, M., and Phillips, B. (2002). Development of the Zebrafish Inner Ear. Developmental Dynamics, 223, 427–458. https://doi.org/10.1002/dvdy.10073

Williams, R. P., Ferlas, B. W., Morales, P. C., and Kurtzweil, A. J. (2017). Mannitol for the prevention of cisplatin-induced nephrotoxicity: A retrospective comparison of hydration plus mannitol versus hydration alone in inpatient and outpatient regimens at a large academic medical center. J Oncol Pharm Practice, 23, 422–428. https://doi.org/10.1177/1078155216656927

Wisniowiecki, A. M., Mattison, S. P., Kim, S., Riley, B., and Applegate, B. E. (2016). Use of a highly transparent zebrafish mutant for investigations in the development of the vertebrate auditory system (Conference Presentation). In Proc.SPIE (Vol. 9716). https://doi.org/10.1117/12.2213087